# ExpWeaver: LLM Agents Learn from Experience via Latent RAG

**Tao Feng** [1]  **Tianyang Luo** [1]  **Jingjun Xu** [1]  **Zhigang Hua** [2]  **Yan Xie** [2]  **Shuang Yang** [2]  **Ge Liu** [1]  **Jiaxuan You** [1]

## Abstract

Experience learning has achieved promising results in enhancing LLM agent planning and reasoning by integrating past interactions as reusable knowledge. However, existing methods remain confined to explicit text space—retrieving experiences via semantic similarity and concatenating them into the context window, leading to substantial token overhead and a decoupled architecture that separates retrieval from generation. To address these limitations, we propose `ExpWeaver`, a framework that enables LLM agents to learn from experience via latent retrieval-augmented generation, without requiring a separate RAG module. `ExpWeaver` encodes experiences using the LLM's own hidden states, retrieves relevant experiences directly in latent space at each decoding step, and integrates them through cross-attention aggregation and gated residual mechanisms. The entire pipeline is optimized end-to-end with reinforcement learning, supporting both generative and ranking tasks. We evaluate `ExpWeaver` on 13 diverse tasks spanning question answering, reasoning, coding, scientific prediction, and recommendation. Results demonstrate that: (1) `ExpWeaver` achieves state-of-the-art on 12 out of 13 tasks, outperforming the strongest baseline by over 6.8%; (2) `ExpWeaver` maintains token efficiency comparable to non-retrieval baselines while text-based retrieval methods require 1.5–2× more tokens; and (3) `ExpWeaver` exhibits superior cross-domain generalization, outperforming the strongest baseline by 16.32% under zero-shot transfer and 15.21% under few-shot transfer. Our code for `ExpWeaver` is released at https://github.com/ulab-uiuc/ExpWeaver.

[1]Department of Computer Science, University of Illinois Urbana-Champaign, Urbana, IL, USA [2]Meta Monetization AI, USA. Correspondence to: Tao Feng <taofeng2@illinois.edu>, Jiaxuan You <jiaxuan@illinois.edu>.

*Proceedings of the $43^{rd}$ International Conference on Machine Learning*, Seoul, South Korea. PMLR 306, 2026. Copyright 2026 by the author(s).

## 1. Introduction

Integrating past interactions as reusable experiences has emerged as a crucial approach for enhancing LLM agent planning and reasoning capabilities (Wang et al., 2023; Shinn et al., 2023; Zhao et al., 2024; Park et al., 2023). Despite this importance, most existing experience learning methods remain confined to explicit text space—storing experiences as textual summaries and retrieving them via text-based semantic similarity (Ouyang et al., 2025; Zhao et al., 2024; Wang et al., 2024). While intuitive, this text-level paradigm introduces inherent inefficiencies and representational bottlenecks that limit scalability and adaptability as tasks grow in complexity. These limitations highlight the need for a more efficient and unified framework for experience learning. Therefore, our paper aims to address this pressing research question: *How can we enable LLM agents to efficiently learn from experience beyond the constraints of explicit text-level operations?*

Existing experience learning approaches can be categorized into retrieval-centric and LLM-centric methods as shown in Table 1. Retrieval-centric methods (Ouyang et al., 2025; Zhao et al., 2024; Wang et al., 2024) focus on optimizing the retriever to select better experiences while keeping the LLM frozen, typically employing preference learning over candidate sets based on LLM-induced utility. This paradigm suffers from the decoupled architecture where the retriever is trained separately from the LLM, preventing joint optimization toward the downstream task objective. LLM-centric methods (Trivedi et al., 2023; Li et al., 2025; Jin et al., 2025) extend flexibility by treating retrieval as a tool-use decision, where the LLM learns when to invoke an external retrieval module through reinforcement learning. While enabling more dynamic retrieval patterns, these methods still rely on a separate RAG module (e.g., search engines or embedding-based retrievers) that operates independently from the LLM's internal representations. These limitations point to the need for an integrated framework that unifies retrieval and generation within a shared representational space.

To address these limitations, we reformulate experience learning as a latent space retrieval-augmented generation problem, where experiences are encoded as dense representations and integrated directly into the LLM's hidden states.

*Table 1.* **Comparison with existing experience learning methods from four perspectives.** Unlike existing methods that require a separate RAG module, `ExpWeaver` performs retrieval directly in the LLM's latent space, enabling adaptive retrieval and support for both generative and ranking tasks.

| Method | Operation Space | Separate RAG | Adaptive Retrieval | Task Type |
|---|---|---|---|---|
| ***Retrieval-Centric Methods*** | | | | |
| ReasoningBank (Ouyang et al., 2025) | Text | ✓ | ✗ | Gen. |
| ExpeL (Zhao et al., 2024) | Text | ✓ | ✗ | Gen. |
| LLM-R (Wang et al., 2024) | Text | ✓ | ✗ | Gen. |
| ***LLM-Centric Methods*** | | | | |
| IRCoT (Trivedi et al., 2023) | Text | ✓ | ✗ | Gen. |
| Search-o1 (Li et al., 2025) | Text | ✓ | ✗ | Gen. |
| Search-R1 (Jin et al., 2025) | Text | ✓ | ✗ | Gen. |
| ExpWeaver | Latent | ✗ | ✓ | Gen. & Rank. |

This shift is important because latent experience integration can bypass context window constraints, enable learnable retrieval criteria, and unify retrieval with generation in a shared representational space. Yet, building an effective latent experience learning framework is far from trivial and comes with several challenges. *First, the embedding space for experiences and queries must remain aligned as the policy evolves during training.* Unlike static retrieval systems, experience learning involves a continuously updating policy, meaning that experiences collected under older policies may become misaligned with the current query representations. Maintaining coherent retrieval without a separate retriever model requires careful design. *Second, aggregating multiple retrieved experiences into a single useful signal is nontrivial.* Different experiences may contain complementary or even conflicting information, and naively pooling them can introduce noise. The aggregation mechanism must learn to weigh experiences based on their relevance to the current generation state. *Third, integrating latent experiences into the generation process requires balancing preservation of the original hidden states with incorporation of external knowledge.* Excessive experience influence may disrupt the model's reasoning flow, while insufficient integration renders the retrieval ineffective. This creates a delicate trade-off that must be learned adaptively.

To tackle the above challenges, we propose `ExpWeaver`, a framework that enables LLM agents to learn from experience via latent retrieval-augmented generation. Specifically, `ExpWeaver` maintains an experience bank where each trajectory is encoded using the *current* policy's hidden states, ensuring on-policy alignment—as the policy evolves, recent experiences naturally reside in a more compatible embedding region, analogous to recency bias in on-policy reinforcement learning (Shao et al., 2024). Further, `ExpWeaver` performs latent retrieval at each decoding step, constructing query embeddings from the current hidden state and selecting experiences via learned similarity. The retrieved experiences are aggregated through cross-attention with a learnable query token, enabling adaptive weighting based on relevance. Finally, we introduce a gated residual mechanism that controls the integration of

aggregated experiences into hidden states, with the mixing coefficients jointly optimized alongside the policy through reinforcement learning. The proposed framework naturally supports diverse task types: for generative tasks such as question answering, reasoning, and coding, the agent produces textual outputs via experience-enhanced decoding; for ranking tasks such as recommendation, the agent computes relevance scores over candidates using the final hidden state representation.

We evaluate `ExpWeaver` on the Experience-driven Benchmark (ExpBench) spanning 13 tasks across 3 scenarios: ExpBench-Generic (question answering, reasoning, and coding), ExpBench-Sci (pharmaceutical chemistry), and ExpBench-Rec (movie and music recommendation). `ExpWeaver` achieves state-of-the-art results on 12 out of 13 tasks, outperforming the strongest baseline by over 6.8% on average. `ExpWeaver` maintains token efficiency comparable to non-retrieval baselines—while text-based methods like Search-R1 require 1.5–2× more tokens due to explicit concatenation. Furthermore, `ExpWeaver` demonstrates strong generalization, outperforming the strongest baseline by 16.32% under zero-shot cross-domain transfer and 15.21% under few-shot transfer, validating that latent experience representations capture more transferable problem-solving patterns.

## 2. Preliminaries

Integrating past interactions as reusable experiences is crucial for enhancing LLM agent planning (Wang et al., 2023; Shinn et al., 2023; Zhao et al., 2024; Park et al., 2023). For each query $x$, the agent generates an output $y$ with a reasoning trace $z$ and receives feedback $r(x, y, z)$. We compress the rollout $\tau = (x, z, y, r)$ into a condensed experience $e$:

$$e = \mathcal{S}(x, y, z, r), \tag{1}$$

where $\mathcal{S}$ is a compression function (e.g., an LLM-based summarizer). A retriever $f_\phi(x, e)$ quantifies the relevance between the current query $x$ and stored experience $e$, constructing a size-$K$ candidate set:

$$\mathcal{C}_\phi(x) = \arg \max_{\mathcal{C} \subseteq \{e\}, |\mathcal{C}|=K} \sum_{e \in \mathcal{C}} f_\phi(x, e). \tag{2}$$

The agent then generates its response: $y \sim p_\theta(y \mid x, \mathcal{C}_\phi(x))$.

**Retrieval-centric Experience Learning**. This paradigm (Ouyang et al., 2025; Zhao et al., 2024; Wang et al., 2024) optimizes the retriever $\phi$ while keeping the LLM $\theta$ fixed. Using an LLM-induced utility $U_\theta(x, \mathcal{C})$ to evaluate candidate sets, the objective is:

$$\max_\phi \mathbb{E}_{x \sim \mathcal{D}} \left[ U_\theta\big(x, \mathcal{C}_\phi(x)\big) \right]. \tag{3}$$

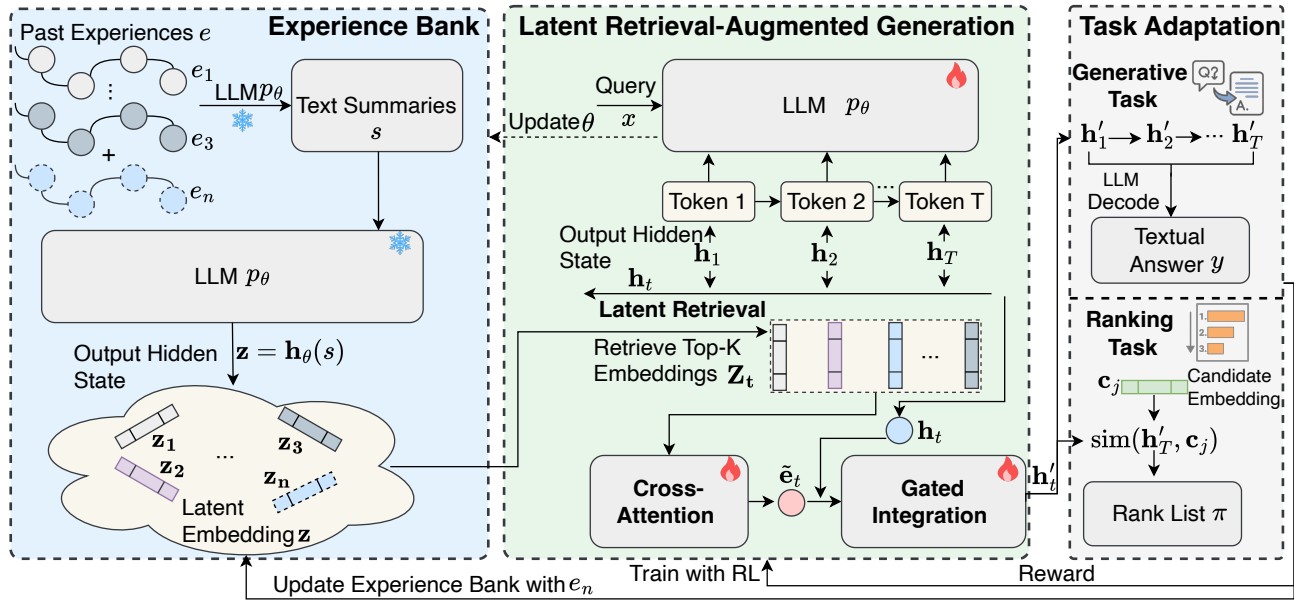

*Figure 1.* **Overview of `ExpWeaver`.** The framework consists of three components: (1) **Experience Bank**: past experiences are summarized and encoded into latent embeddings $\mathbf{z} = \mathbf{h}_\theta(s)$ using the same LLM $p_\theta$; (2) **Latent Retrieval-Augmented Generation**: at each decoding step, the hidden state $\mathbf{h}_t$ retrieves Top-K experience embeddings $\mathbf{Z}_t$, which are aggregated via cross-attention and integrated through a gated mechanism to produce $\mathbf{h}'_t$; (3) **Task Adaptation**: the enhanced hidden states support both generative tasks (producing textual output $y$) and ranking tasks (computing $\mathrm{sim}(\mathbf{h}'_T, \mathbf{c}_j)$ for candidate ranking). The entire pipeline is trained end-to-end with reinforcement learning. Snowflake denotes frozen parameters; flame denotes trainable parameters.

However, this paradigm faces key limitations: *(i)* text-level concatenation consumes substantial context window, limiting retrievable experiences; *(ii)* retrieval relies on semantic similarity, yet similar experiences are not necessarily beneficial for the query.

**LLM-centric Experience Learning**. This paradigm (Trivedi et al., 2023; Li et al., 2025; Jin et al., 2025) treats retrieval as a tool-use decision, directly optimizing $p_\theta$ for end-to-end performance. Let $\pi_\theta(\mathcal{C} \mid x)$ denote the model-induced distribution over the candidate sets. The agent samples $\mathcal{C} \sim \pi_\theta(\cdot \mid x)$, generates $y \sim p_\theta(y \mid x, \mathcal{C})$, and maximizes:

$$\max_\theta \ \mathbb{E}_{x \sim \mathcal{D}} \left[ \mathbb{E}_{\mathcal{C} \sim \pi_\theta(\cdot \mid x)} \mathbb{E}_{y \sim p_\theta(\cdot \mid x, \mathcal{C})} \left[ r(x, y) \right] \right]. \quad (4)$$

While this couples retrieval with generation, they remain architecturally separated, preventing adaptive learning of what experiences the agent truly needs. These limitations motivate an integrated framework unifying retrieval and generation within a shared representational space.

## 3. `ExpWeaver`: Learning from Experience with Latent RAG

Figure 1 illustrates the overall architecture of `ExpWeaver`. Given a query, the agent leverages the past experiences stored in the experience bank to generate a response. Unlike

text-based methods that concatenate experiences into the context window, `ExpWeaver` operates entirely in latent space: experiences are encoded as dense embeddings using the LLM's own hidden states, retrieved at each decoding step based on similarity with the current hidden state, aggregated via cross-attention, and integrated through a gated mechanism to produce enhanced representations. The framework naturally supports both generative tasks and ranking tasks, with all parameters jointly optimized via reinforcement learning.

### 3.1. Experience Bank

**Experience Representation**. We maintain an experience bank $\mathcal{M}$ that stores the agent's problem-solving trajectories as reusable experiences. Each experience $e \in \mathcal{M}$ is represented as:

$$e = (x, z, y, y^\star, r, s, \mathbf{z}), \quad (5)$$

where $x$ is the query, $z$ is the reasoning trace, $y$ is the generated output, $y^\star$ is the reference answer, $r$ is the reward, $s = \mathcal{S}(x, y, z, r)$ is a textual summary produced by prompting the LLM $p_\theta$, and $\mathbf{z} \in \mathbb{R}^d$ is its latent embedding with $d$ being the hidden dimension.

A key design choice is to compute the embedding using the *same LLM* used for generation rather than a separate

**Algorithm 1** Training of `ExpWeaver`

---

**Require:** Dataset $\mathcal{D} = \{(x, y^\star)\}$, policy $p_\theta$, experience parameters $\psi_{\mathrm{exp}}$, group size $G$, retrieval size $K$, learning rate $\eta$, KL coefficient $\beta$, cold-start threshold $M_{\min}$.

1: Initialize experience bank $\mathcal{M} \leftarrow \emptyset$
2: **for** each iteration **do**
3:     Sample mini-batch $\mathcal{B} \sim \mathcal{D}$
4:     **for** each $(x, y^\star) \in \mathcal{B}$ **do**
5:         **for** $i = 1$ to $G$ **do**
6:             **if** $|\mathcal{M}| < M_{\min}$ **then**
7:                 Generate $(y_i, z_i) \sim p_\theta(\cdot \mid x)$    ▷ Cold-start: skip retrieval
8:             **else**
9:                 Generate $(y_i, z_i) \sim p_\theta(\cdot \mid x, \mathcal{C})$   ▷ Eq. 8-12
10:             **end if**
11:             Compute reward $r_i \leftarrow r(x, y_i)$
12:         **end for**
13:         **for** $i = 1$ to $G$ **do**
14:             Summarize $s_i \leftarrow \mathcal{S}(x, y_i, z_i, r_i)$
15:             Encode $\mathbf{z}_i \leftarrow \mathbf{h}_\theta(s_i)$      ▷ Eq. 6
16:             $\mathcal{M} \leftarrow \mathcal{M} \cup \{(x, z_i, y_i, y^\star, r_i, s_i, \mathbf{z}_i)\}$
17:         **end for**
18:         Compute $\hat{A}_i \leftarrow (r_i - \bar{r})/(\sigma_r + \epsilon)$ for all $i$
19:     **end for**
20:     Compute $\mathcal{L}(\theta, \psi_{\mathrm{exp}})$          ▷ Eq. 13
21:     $(\theta, \psi_{\mathrm{exp}}) \leftarrow (\theta, \psi_{\mathrm{exp}}) - \eta \nabla \mathcal{L}(\theta, \psi_{\mathrm{exp}})$
22: **end for**

---

retriever:

$$\mathbf{z} = \mathbf{h}_\theta(s), \tag{6}$$

where $\mathbf{h}_\theta(s)$ denotes the hidden state of the last token at the final transformer layer of $p_\theta$ when processing $s$. This unified approach eliminates the need for a dedicated retrieval model and ensures *on-policy alignment*—as $\theta$ evolves during training, recent experiences naturally reside in a more compatible embedding region, creating an implicit recency bias analogous to on-policy reinforcement learning (Shao et al., 2024).

**Storage and Indexing.** We build an index over $L_2$-normalized embeddings with cosine similarity:

$$\mathrm{sim}(\mathbf{z}, \mathbf{z}') = \frac{\mathbf{z}^\top \mathbf{z}'}{\|\mathbf{z}\| \, \|\mathbf{z}'\|}. \tag{7}$$

The bank enforces a fixed capacity by evicting the oldest entries when full.

### 3.2. Latent Retrieval-Augmented Generation

We now describe how experiences are retrieved and integrated during autoregressive generation $y \sim p_\theta(y \mid x, \mathcal{C})$, where $\mathcal{C}$ denotes the retrieved candidate set. Let $\mathbf{h}_t \in \mathbb{R}^d$

denote the hidden state at the $t$-th decoding step at the final transformer layer of $p_\theta$.

**Latent Experience Retrieval.** We retrieve experiences directly in the LLM's latent space at each decoding step. Specifically, we use the current hidden state $\mathbf{h}_t$ as the query and retrieve a size-$K$ candidate set:

$$\mathcal{C}_t = \mathrm{TopK}_{e \in \mathcal{M}}\big(\mathrm{sim}(\mathbf{h}_t, \mathbf{z}_e), K\big), \tag{8}$$

where $\mathbf{z}_e$ is the embedding of experience $e$. Since both $\mathbf{h}_t$ and $\mathbf{z}_e$ are derived from the same LLM $p_\theta$, no separate retriever is required.

**Cross-Attention Aggregation.** Given $\mathcal{C}_t = \{e_1, \dots, e_K\}$, we stack their embeddings into a matrix $\mathbf{Z}_t = [\mathbf{z}_{e_1}; \dots; \mathbf{z}_{e_K}] \in \mathbb{R}^{K \times d}$ and aggregate via cross-attention:

$$\mathbf{e}_t = \mathrm{LN}\big(\mathrm{CrossAttn}(\mathbf{u}, \mathbf{Z}_t, \mathbf{Z}_t)\big), \tag{9}$$

where $\mathbf{u} \in \mathbb{R}^d$ is a learnable query token and LN denotes layer normalization. We rescale to match the hidden state magnitude: $\tilde{\mathbf{e}}_t = \mathbf{e}_t \cdot \|\mathbf{h}_t\|/\|\mathbf{e}_t\|$.

**Gated Experience Integration.** We integrate the aggregated experience through a gated residual mechanism with retention and input gates:

$$\mathbf{r}_t = \sigma(\mathbf{W}_r \mathbf{h}_t), \quad \mathbf{i}_t = \sigma(\mathbf{W}_i \mathbf{h}_t), \tag{10}$$

where $\mathbf{W}_r, \mathbf{W}_i \in \mathbb{R}^{d \times d}$ are learnable and $\sigma$ is the sigmoid function. The mixing coefficient is:

$$\mathbf{a}_t = \exp\big(-\alpha \cdot \mathrm{softplus}(-\boldsymbol{\lambda}) \odot \mathbf{r}_t\big), \tag{11}$$

where $\boldsymbol{\lambda} \in \mathbb{R}^d$ is learnable and $\alpha > 0$ is a scaling hyperparameter. The updated hidden state is computed via norm-preserving interpolation:

$$\mathbf{h}'_t = \mathbf{a}_t \odot \mathbf{h}_t + \sqrt{1 - \mathbf{a}_t^2} \odot (\mathbf{i}_t \odot \tilde{\mathbf{e}}_t). \tag{12}$$

All additional parameters $\psi_{\mathrm{exp}} = \{\mathbf{u}, \mathbf{W}_r, \mathbf{W}_i, \boldsymbol{\lambda}\}$ are jointly optimized with $\theta$ during training.

**Task Adaptation.** The proposed framework naturally extends to different task types. For *generative tasks* (question answering, reasoning, coding, and scientific prediction), the enhanced hidden states $\{\mathbf{h}'_t\}$ are used for next-token prediction, producing a textual output $y \sim p_\theta(y \mid x, \mathcal{C})$. For *ranking tasks* with candidate set $\mathcal{Y} = \{y_1, \dots, y_M\}$ given user context $x$, we encode each candidate as $\mathbf{c}_j = \mathbf{h}_\theta(y_j)$ using the same LLM, perform a forward pass with latent retrieval, and compute the relevance score $\mathrm{score}(y_j \mid x) = \mathrm{sim}(\mathbf{h}'_T, \mathbf{c}_j)$, where $\mathbf{h}'_T$ is the final hidden state. The ranking is produced by sorting candidates in descending order.

### 3.3. Training with Reinforcement Learning

**Reward Design.** We design reward functions based on the task output type. For *generative tasks*, the reward

$r_{\text{gen}}(y, y^\star) = \mathcal{F}(y, y^\star)$ measures correspondence between output $y$ and ground truth $y^\star$, where $\mathcal{F}$ is a task-appropriate function (e.g., exact matching for QA, pass@1 for coding). For *ranking tasks*, let $\rho_\pi(y^\star) \in \{1, \ldots, M\}$ denote the position of ground-truth item $y^\star$ in ranking $\pi$. The reward $r_{\text{rank}}(\pi, y^\star) = 1/\rho_\pi(y^\star)$ yields 1 when ranked first and decreases as position lowers.

**Policy Optimization**. We optimize with Group Relative Policy Optimization (GRPO) (Shao et al., 2024). For each query $x$, we sample $G$ responses $\{y_i\}_{i=1}^G$ with rewards $\{r_i\}_{i=1}^G$. The objective is:

$$
\mathcal{L}(\theta, \psi_{\text{exp}}) = -\mathbb{E}_{x \sim \mathcal{D}} \left[ \frac{1}{G} \sum_{i=1}^G \frac{p_\theta(y_i \mid x)}{p_{\theta_{\text{old}}}(y_i \mid x)} \hat{A}_i \right]
$$
$$
+ \beta \, \mathbb{E}_{x \sim \mathcal{D}} \left[ \frac{1}{G} \sum_{i=1}^G D_{\text{KL}}(p_\theta \| p_{\text{ref}}) \right], \quad (13)
$$

where $\hat{A}_i = (r_i - \bar{r})/(\sigma_r + \epsilon)$ is the group-normalized advantage and $\beta$ controls KL regularization. The training procedure is summarized in Algorithm 1.

# 4. Experiments

To explore the capability of LLM agents for transforming past interactions into reusable knowledge via latent retrieval, we conduct a comprehensive training and evaluation of the proposed `ExpWeaver` across 13 interdisciplinary decision-making tasks in the Experience-driven Benchmark (ExpBench) with varying environmental complexities and decision-making horizons. We then compare its performance against both general reasoning-based baselines and state-of-the-art experience-learning methods, including retrieval-centric experience learning methods and LLM-centric experience learning methods. We begin by introducing the task environments and evaluation metrics within the experience-augmented agent framework.

**Task description**. The details of the tasks are summarized across three scenarios in Table 24 of the Appendix. **(i) ExpBench-Generic.** To evaluate the generalization of experience learning across diverse generic LLM tasks, we employ ten widely used benchmarks spanning three categories as shown in Table 24: (1) *Question Answering* (ARC-C (Clark et al., 2018b), CommonsenseQA (Talmor et al., 2019b), GPQA (Rein et al., 2024), MMLU (Hendrycks et al., 2020), OBQA (Mihaylov et al., 2018b)), which assess commonsense and multi-domain knowledge with accuracy as the metric; (2) *Mathematical Reasoning* (GSM8K (Cobbe et al., 2021b), GSM-Symbolic (Mirzadeh et al., 2024), MATH (Hendrycks et al., 2021b)), which evaluate numerical and symbolic reasoning with exact match (Rajpurkar et al., 2016); (3) *Code Generation* (HumanEval+ and MBPP+ (Liu et al., 2023b)), which test program synthesis

with Pass@1 (Chen, 2021); **(ii) ExpBench-Sci.** To investigate whether experience learning can effectively transfer domain-specific scientific knowledge, we adopt the Chem-TDC (Therapeutics Data Commons) (Huang et al., 2021b), which contains pharmaceutical chemistry questions from the TDC that involve expert-level chemical knowledge not captured by general reasoning alone. We use exact match as the evaluation metric; **(iii) ExpBench-Rec.** To examine the applicability of experience learning in ranking-based decision scenarios, we employ two sequential recommendation datasets: Rec-Movie (MovieLens ml-1m) (Harper & Konstan, 2015) and Rec-Music (Ni et al., 2019a). Following prior studies (Geng et al., 2022; Hua et al., 2023; Feng et al., 2025), we construct each sample by extracting 20 consecutive interactions as the historical sequence, while designating the 21st interaction as the ground-truth item. For evaluation, we adopt the widely used leave-one-out strategy and report Normalized Discounted Cumulative Gain (NDCG@10) (Järvelin & Kekäläinen, 2002; Burges et al., 2005) and Mean Reciprocal Rank (MRR) (Voorhees et al., 1999; Cremonesi et al., 2010) as ranking metrics.

**Baselines**. We evaluate a variety of baseline methods across three scenarios. The baselines are categorized into three groups: **(a)** *General Reasoning-Based Baselines* that apply LLM reasoning across tasks without leveraging external experience retrieval; **(b)** *Retrieval-Centric Experience Learning Baselines* that focus on optimizing the retriever parameters while keeping the LLM fixed; **(c)** *LLM-Centric Experience Learning Baselines* that treat retrieval as an external tool-use decision made by the LLM itself. **(i) General Reasoning-Based Baselines.** We consider methods that apply LLM reasoning across tasks without leveraging external experience retrieval: 1) *CoT* (Wei et al., 2022): Chain-of-Thought prompting that elicits step-by-step reasoning from LLMs to improve performance on complex tasks. 2) *HRPO* (Zheng et al., 2025): A reasoning optimization method that enhances LLM performance through hierarchical preference learning. 3) *R1* (Guo et al., 2025): A strong reasoning model that applies reinforcement learning for multi-step reasoning. Particularly for ExpBench-Rec, we additionally include two recent RL-based LLM recommendation baselines, as this scenario has witnessed rapid development in reinforcement learning methods that directly optimize LLMs for recommendation tasks: 4) *Rec-R1* (Lin et al., 2025): A general reinforcement learning framework that bridges LLMs with recommendation systems through closed-loop optimization, directly optimizing LLM generation using feedback from black-box recommendation models via GRPO. 5) *IRanker* (Feng et al., 2025): A unified ranking foundation model that decomposes complex ranking tasks into an iterative decoding process, progressively eliminating the worst candidate step-by-step while leveraging reinforcement learning for optimization; **(ii) Retrieval-Centric Experience Learning**

*Table 2.* **Performance comparison across ExpBench-Generic with 10 diverse tasks.** Results are grouped by task category: Question Answering, Reasoning, and Coding. **Bold** and underline denote the best and second-best results.

| Method | Question Answering | | | | | Reasoning | | | Coding | | Avg. |
|---|---|---|---|---|---|---|---|---|---|---|---|
| | ARC-C | CommonsenseQA | GPQA | MMLU | OBQA | GSM8K | GSM-Symbolic | MATH | HumanEval+ | MBPP+ | |
| *General Reasoning-Based Baselines* | | | | | | | | | | | |
| CoT | 62.44 | 56.44 | 26.67 | 58.22 | 64.22 | 71.11 | 66.44 | 53.11 | 66.00 | 72.81 | 61.40 |
| HRPO | 79.33 | 75.33 | 20.00 | 67.11 | 72.00 | 88.22 | 80.89 | 63.11 | **74.36** | 70.89 | 74.04 |
| R1 | 78.44 | 74.67 | 21.67 | 66.22 | 71.33 | 87.11 | 79.78 | 61.78 | 73.08 | 68.75 | 73.10 |
| *Retrieval-Centric Experience Learning Baselines* | | | | | | | | | | | |
| ReasoningBank | 74.89 | 63.33 | 30.00 | 60.44 | 67.56 | 68.00 | 63.11 | 32.89 | 52.00 | 52.63 | 60.57 |
| ExpeL | 76.44 | 66.44 | 26.67 | 67.54 | 69.56 | 61.33 | 58.89 | 36.00 | 60.00 | 67.54 | 60.94 |
| LLM-R | 74.89 | 67.56 | 16.67 | 61.33 | 71.78 | 73.78 | 65.11 | 35.33 | 64.00 | 67.54 | 63.52 |
| *LLM-Centric Experience Learning Baselines* | | | | | | | | | | | |
| IRCoT | 73.56 | 64.22 | 33.33 | 60.22 | 68.89 | 65.11 | 63.11 | 28.67 | 52.00 | 55.26 | 59.82 |
| Search-o1 | 80.00 | 72.00 | 28.33 | 63.78 | 71.56 | 66.89 | 63.11 | 30.89 | 54.00 | 57.02 | 63.10 |
| Search-R1 | 80.44 | 76.67 | 23.33 | 67.78 | 73.33 | 85.11 | 77.56 | 59.78 | 71.79 | 67.50 | 73.27 |
| ExpWeaver | **84.00** | **82.00** | **35.00** | **68.00** | **80.00** | **89.78** | **85.00** | **65.33** | 72.00 | **78.00** | **78.25** |

**Baselines.** These methods focus on optimizing the retriever while keeping the LLM fixed, using LLM-induced utility for preference learning over candidate sets: 1) *ReasoningBank* (Ouyang et al., 2025): A retrieval-augmented reasoning method that maintains a bank of reasoning examples for retrieval during inference. 2) *ExpeL* (Zhao et al., 2024): An experience learning framework that extracts and retrieves task-specific experiences via KNN to enhance LLM reasoning. 3) *LLM-R* (Wang et al., 2024): A retrieval-centric method that learns to retrieve in-context examples for large language models via pairwise ranking loss; **(iii) LLM-Centric Experience Learning Baselines.** These methods treat retrieval as a tool-use decision and directly optimize the LLM for end-to-end performance: 1) *IRCoT* (Trivedi et al., 2023): An interleaved retrieval chain-of-thought method that dynamically retrieves information during reasoning steps. 2) *Search-o1* (Li et al., 2025): A search-augmented reasoning framework that integrates agentic search capabilities into the reasoning process. 3) *Search-R1* (Jin et al., 2025): A reinforcement learning-based method that trains LLMs to effectively utilize search engines for improved reasoning.

**Implementation Details**. We implement ExpWeaver based on Qwen2.5-3B-Instruct[1] using the Unsloth library[2] for efficient training. We apply LoRA (Hu et al., 2022) with rank 32 and scaling factor 64 to all attention and feed-forward layers. The experience bank is indexed by FAISS (Johnson et al., 2019) with inner product search on $L_2$-normalized embeddings (Eq. 7) and retrieval num $K = 3$; we randomly sample one trajectory per group to add to $\mathcal{M}$ during training. For cross-attention aggregation

(Eq. 9), we use 8 attention heads with dropout rate 0.1. The learnable parameter $\boldsymbol{\lambda}$ (Eq. 11) is initialized such that $\mathbf{a}_t \in [0.980, 0.999]$, ensuring the model primarily relies on its hidden states $\mathbf{h}_t$ early in training. We use GRPO (Eq. 13) (Shao et al., 2024) implemented via TRL[3] with group size $G = 4$, KL coefficient $\beta = 0.005$, and gradient clipping with max norm 0.1. The learning rate is $5 \times 10^{-6}$ for LoRA parameters and $1 \times 10^{-4}$ for $\psi_{\text{exp}}$, following cosine decay with 10% warmup, using paged AdamW (Loshchilov & Hutter, 2017) with weight decay 0.1. We set maximum prompt and completion lengths to 1024 tokens each, with per-device batch size 8 and gradient accumulation over 4 steps (effective batch size 32), enabling BF16 mixed precision and gradient checkpointing. During training, we sample with temperature 0.7; for evaluation, we use greedy decoding. All experiments are conducted on 4 NVIDIA A6000 GPUs.

## 4.1. ExpWeaver Outperforms General Reasoning-Based and Experience Learning Methods

We evaluate ExpWeaver on three ExpBench suites: Generic, Sci, and Rec. We train a separate model for each suite while keeping all hyperparameters consistent across experiments. Results are reported in Table 2, Table 3, and Table 4. We have the following observations.

**ExpWeaver Attains SOTA Results Across Diverse Scenarios**. ExpWeaver achieves SOTA results on 12 out of 13 tasks with over 6.8% improvement compared to the strongest baseline in each scenario, demonstrating robustness across diverse task types. We analyze why

---

[1] https://huggingface.co/Qwen/Qwen2.5-3B-Instruct

[2] https://github.com/unslothai/unsloth

[3] https://github.com/huggingface/trl

*Table 3.* **Model performance comparison on the ExpBench-Sci.** **Bold** and underline denote the best and second-best results.

| Method | Chem-TDC |
|---|---|
| *General Reasoning-Based Baselines* | |
| CoT | 57.60 |
| HRPO | 60.22 |
| R1 | 62.22 |
| *Retrieval-Centric Experience Learning Baselines* | |
| ReasoningBank | 57.80 |
| ExpeL | 58.44 |
| LLM-R | 59.11 |
| *LLM-Centric Experience Learning Baselines* | |
| IRCoT | 54.20 |
| Search-o1 | 55.10 |
| Search-R1 | 63.78 |
| ExpWeaver | **69.58** |

*Table 4.* **Model performance comparison on ExpBench-Rec containing 2 tasks using NDCG@10 and MRR metrics. Bold** and underline denote the best and second-best results.

| | Rec-Movie | | Rec-Music | | Avg. | |
|---|---|---|---|---|---|---|
| Method | NDCG@10 | MRR | NDCG@10 | MRR | NDCG@10 | MRR |
| *General Reasoning-Based Baselines* | | | | | | |
| CoT | 19.72 | 15.31 | 23.58 | 19.65 | 21.65 | 17.48 |
| HRPO | 21.50 | 16.80 | 24.50 | 19.80 | 23.00 | 18.30 |
| Rec-r1 | 23.98 | 18.71 | 20.50 | 15.70 | 22.24 | 17.21 |
| IRanker | 42.32 | 34.69 | 33.47 | 29.18 | 37.90 | 31.94 |
| *Retrieval-Centric Experience Learning Baselines* | | | | | | |
| ReasoningBank | 17.82 | 13.88 | 23.92 | 19.68 | 20.87 | 16.78 |
| ExpeL | 19.31 | 15.69 | 25.27 | 20.60 | 22.29 | 18.15 |
| LLM-R | 19.81 | 16.02 | 25.43 | 21.09 | 22.64 | 18.56 |
| *LLM-Centric Experience Learning Baselines* | | | | | | |
| IRCoT | 19.42 | 15.18 | 22.88 | 18.62 | 21.15 | 16.90 |
| Search-o1 | 19.88 | 15.58 | 22.72 | 18.62 | 21.30 | 17.10 |
| Search-R1 | 26.34 | 19.55 | 30.72 | 25.33 | 28.53 | 22.44 |
| ExpWeaver | **49.37** | **39.55** | **39.42** | **33.21** | **44.40** | **36.38** |

ExpWeaver outperforms Search-R1, the strongest LLM-centric baseline. As shown in Table 2, Search-R1 performs competitively on QA tasks but degrades on Reasoning and Coding. This is because Search-R1 relies on text-level retrieval based on surface semantic similarity, which works for knowledge-intensive QA where similar experiences contain transferable information. However, reasoning and coding require capturing deeper structural correspondences in problem-solving strategies rather than superficial textual overlap. ExpWeaver addresses this through latent space retrieval and cross-attention aggregation, learning query-dependent experience utilization that captures these deeper connections. This explains ExpWeaver's consistent gains across all categories, including ExpBench-Sci (Table 3) and ExpBench-Rec (Table 4).

**ExpWeaver Achieves Superior Performance with Lower Token Cost**. Beyond accuracy improvements, ExpWeaver also demonstrates remarkable efficiency in token consumption. As illustrated in Figure 2, Search-R1

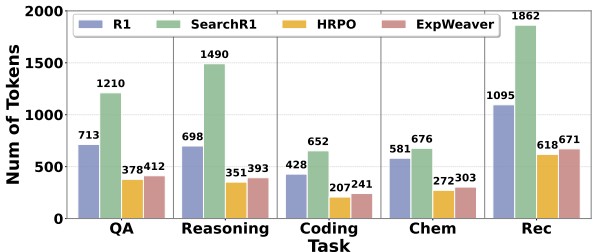

*Figure 2.* **ExpWeaver achieves superior performance with significantly lower token consumption.** We compare the average number of tokens consumed per query for R1, Search-R1, HRPO, and ExpWeaver across five task categories. Search-R1 exhibits the highest token consumption due to explicit text-level experience concatenation, while ExpWeaver maintains comparable token efficiency to HRPO by compressing experiences into latent representations, thereby avoiding context window overhead.

exhibits the highest token consumption across all task categories, requiring approximately $1.5\times$ to $2\times$ more tokens than R1 due to explicit text-level experience concatenation. This overhead becomes particularly severe in Reasoning and Rec scenarios. In contrast, ExpWeaver achieves token efficiency comparable to HRPO while substantially outperforming it in accuracy. Specifically, ExpWeaver consumes only approximately 10% more tokens than HRPO, yet delivers over 4% improvement in overall performance. This efficiency stems from ExpWeaver's latent experience integration: rather than concatenating raw experience text, ExpWeaver compresses experiences into dense latent representations and integrates them via cross-attention, eliminating context window overhead while preserving informational value. These results demonstrate that ExpWeaver achieves the best of both worlds—superior performance and computational efficiency.

### 4.2. ExpWeaver Exhibits Superior Generalization Capabilities Across Different Tasks

To evaluate the generalization capability of experience-based methods, we conduct cross-domain transfer experiments on ExpBench-Sci (Chem-TDC). As illustrated in Figure 3, we compare ExpWeaver and Search-R1 under three settings: ZeroShot (model and experience bank trained on ExpBench-Generic, directly evaluated on Chem-TDC), FewShot (experience bank from ExpBench-Generic, model fine-tuned on Chem-TDC), and InDomain (both trained on Chem-TDC). We include R1-InDomain and R1-ZeroShot as reference baselines.

ExpWeaver demonstrates consistently superior generalization across all settings. In the ZeroShot setting, ExpWeaver outperforms Search-R1 by over 8%, indicating that latent experience representations transfer more effectively across domains than explicit text-based retrieval. In the FewShot setting, ExpWeaver maintains a similar

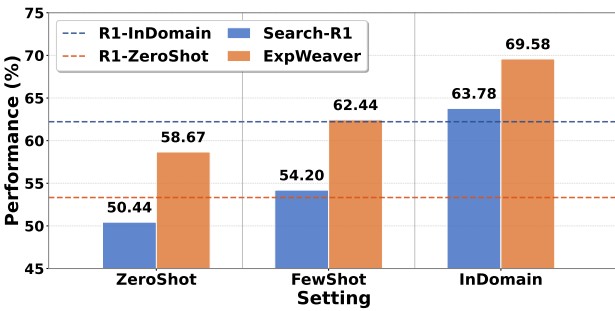

*Figure 3.* **ExpWeaver exhibits superior cross-domain generalization on ExpBench-Sci.** We evaluate generalization capability under three settings: (1) **ZeroShot**: the model and experience bank are trained on ExpBench-Generic (QA, Reasoning, Coding) and directly evaluated on Chem-TDC without any domain-specific adaptation; (2) **FewShot**: the experience bank is from ExpBench-Generic, but the model is fine-tuned on Chem-TDC; (3) **InDomain**: both model and experience bank are trained and evaluated on Chem-TDC. Horizontal dashed lines indicate R1-InDomain and R1-ZeroShot baselines. `ExpWeaver` consistently outperforms Search-R1 across all settings, with particularly notable gains in ZeroShot and FewShot scenarios.

advantage, surpassing Search-R1 by approximately 8%. Notably, `ExpWeaver`-FewShot even exceeds R1-InDomain, demonstrating that cross-domain experiences combined with minimal domain adaptation can match or surpass fully in-domain trained reasoning models.

We further analyze the performance retention relative to the InDomain setting. `ExpWeaver` retains approximately 84% of its InDomain performance under ZeroShot and 90% under FewShot, whereas Search-R1 retains only 79% and 85%, respectively. This smaller performance gap indicates that `ExpWeaver`'s latent experience integration mechanism captures more transferable problem-solving patterns that generalize beyond surface-level domain-specific features. These results validate that `ExpWeaver` not only excels in in-domain scenarios but also exhibits robust cross-domain generalization capabilities.

### 4.3. Ablation Studies Validate `ExpWeaver`'s Key Components

**`ExpWeaver` Shows Robust Performance Across Different Retrieval Numbers**. We vary the retrieval number $K \in \{1, 3, 5, 8\}$ to study its impact on performance. As shown in Figure 4(a), `ExpWeaver` demonstrates robust performance across different $K$ values, with $K=3$ and $K=5$ achieving nearly identical best results. When $K=1$, performance drops noticeably across all tasks, indicating that a single retrieved experience provides insufficient guidance. When $K=8$, performance slightly degrades compared to $K=3$ and $K=5$, likely due to noise introduced by less relevant experiences. These results suggest that `ExpWeaver`

is not overly sensitive to the choice of $K$, and a moderate value ($K=3$) suffices for optimal performance.

**`ExpWeaver` Benefits from Cross-Attention Aggregation**. To assess the cross-attention aggregation mechanism, we design three variants: **(i) Mean Pooling**: replaces cross-attention with average pooling over retrieved embeddings, treating all experiences equally; **(ii) Weighted Mean**: computes weighted average based on retrieval similarity scores, introducing relevance-aware but non-learnable aggregation; **(iii) Qformer**: replaces cross-attention with a Qformer-style (Li et al., 2023) architecture using multiple learnable query tokens and additional transformer layers. As shown in Figure 4(b), Mean Pooling suffers the largest degradation, demonstrating that uniform aggregation fails to distinguish useful experiences. Weighted Mean improves by incorporating retrieval scores, but static weighting cannot adapt to evolving generation context. Qformer achieves competitive results but does not surpass `ExpWeaver` despite increased complexity, suggesting over-parameterization yields diminishing returns. In contrast, `ExpWeaver`'s cross-attention strikes an effective balance: a single learnable query token dynamically weights experiences based on the current decoding state, enabling adaptive integration at each generation step.

**`ExpWeaver` Performs Best with Moderate Mixing Coefficient Range**. We investigate the effect of initializing the mixing coefficient $\mathbf{a}_t$ (Eq. 11), which controls the balance between the original hidden state $\mathbf{h}_t$ and the integrated experience $\tilde{\mathbf{e}}_t$. We denote $\mathbf{a}_{min}$ as the lower bound of the initial range for $\mathbf{a}_t$, with the upper bound fixed at 0.999. We vary $\mathbf{a}_{min} \in \{0.95, 0.97, 0.98, 0.99\}$ to study its impact. As shown in Figure 4(c), $\mathbf{a}_{min}=0.98$ achieves the best overall performance. When $\mathbf{a}_{min}$ is too small (0.95), the model allows excessive experience influence early in training, potentially disrupting the pretrained representations. When $\mathbf{a}_{min}$ is too large (0.99), the narrow initialization range limits the model's flexibility to integrate experiences. The optimal value $\mathbf{a}_{min}=0.98$ strikes a balance, enabling the model to primarily preserve the hidden states initially while gradually learning appropriate experience integration during training.

## 5. Additional Related Work

**Reinforcement Learning for LLMs**. RL has evolved from PPO-based alignment (Ouyang et al., 2022; Liu et al., 2024) to diverse paradigms including Direct Preference Optimization (DPO) (Rafailov et al., 2023), Process Reward Models (PRMs) (Lightman et al., 2023), and self-play frameworks like SPIN (Chen et al., 2024) and SCoRe (Kumar et al., 2024). Recent advances such as GRPO (Shao et al., 2024), Dr.GRPO (Liu et al., 2025), GSPO (Zheng et al., 2025), and Clip-Cov (Cui et al., 2025) further specialize reasoning in mathematical (Shao et al., 2024), competitive (Han et al.,

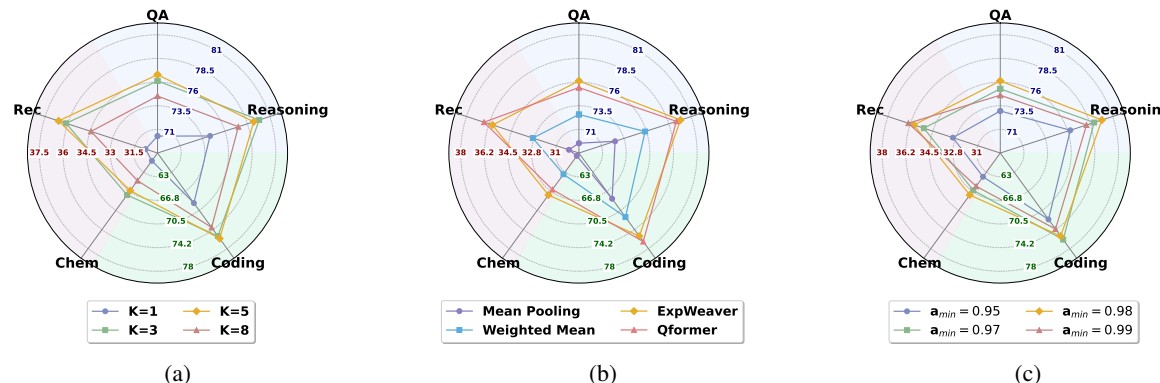

*Figure 4.* **Ablation studies validate `ExpWeaver`'s key components across five task categories.** (a) *Effect of retrieval number $K$:* `ExpWeaver` shows robust performance across different $K$ values, with $K$=3 and $K$=5 achieving the best overall results. (b) *Effect of aggregation mechanism:* Cross-attention aggregation (`ExpWeaver`) consistently outperforms Mean Pooling, Weighted Mean, and Qformer variants, demonstrating the effectiveness of learnable experience aggregation. (c) *Effect of mixing coefficient initialization* $\mathbf{a}_{\min}$: `ExpWeaver` performs best with moderate initialization range ($\mathbf{a}_{\min}$=0.98), balancing experience integration and hidden state preservation.

2025), and temporal (Li et al., 2017) domains. Beyond policy updates, RL training generates exploration trajectories that can be summarized into reusable experiences, distilling knowledge about effective reasoning patterns and failure modes to inform agent planning and improve sample efficiency.

**Experience Learning of LLM Agents**. LLM agents increasingly leverage past interactions to enhance reasoning and planning. One line designs memory systems for intra-task state management: MemGPT (Packer et al., 2023), MemoryBank (Zhong et al., 2024), and CoALA (Sumers et al., 2023) maintain working context within a single episode. **These memory systems address intra-query coordination, tracking how reasoning steps connect within one task. In contrast, experience learning targets inter-query transfer, extracting reusable patterns from past tasks to guide future ones. This paper focuses on the latter.** Along this direction, IRCoT (Trivedi et al., 2023) and Search-o1 (Li et al., 2025) interleave retrieval to refine reasoning at inference, while ReasoningBank (Ouyang et al., 2025) and ExpeL (Zhao et al., 2024) store traces and success-failure signals as reusable repositories. LLM-R (Wang et al., 2024) utilizes task utility for selection and Search-R1 (Jin et al., 2025) employs RL for query-dependent utilization. However, prior methods integrate experiences via explicit text concatenation, limiting end-to-end optimization. This motivates `ExpWeaver`, which compresses rollouts into latent representations for adaptive, differentiable guidance.

## 6. Conclusion

We present `ExpWeaver`, a novel framework that enables LLM agents to learn from experience via latent retrieval-augmented generation. `ExpWeaver` compresses past roll-outs into latent representations and integrates them into the generation process through cross-attention aggregation and gated experience integration. Extensive experiments across 13 tasks demonstrate that `ExpWeaver` achieves state-of-the-art performance on 12 tasks while maintaining comparable token efficiency to non-retrieval baselines. Furthermore, `ExpWeaver` exhibits superior cross-domain generalization, effectively transferring learned experiences to unseen domains.

## Acknowledgments

We sincerely appreciate the support from the research gift from Meta.

## Impact Statement

This paper presents ExpWeaver, a framework designed to enhance the reasoning capabilities of large language models through experience-based learning. Our work aims to advance the field of machine learning by enabling more efficient and effective knowledge utilization across diverse tasks. ExpWeaver demonstrates significant improvements in reasoning, question answering, and recommendation tasks while substantially reducing computational costs measured by token usage. This efficiency gain could increase access to advanced AI capabilities by lowering the resource requirements for deploying reasoning-augmented systems. The cross-domain transferability of learned experiences may also accelerate development in specialized domains such as scientific research and healthcare, where labeled data is scarce. We do not believe this work poses unique ethical concerns beyond those generally associated with large language model research. The techniques presented are general-purpose and domain-agnostic, with no specific ap-

plications that raise immediate societal concerns. We hope that the efficiency improvements offered by ExpWeaver will contribute to more sustainable AI development by reducing computational requirements.

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

## A. Prompt Usage of All Methods

This section provides a detailed overview of the prompt templates used for each task scenario, corresponding to all methods. Each prompt is carefully designed with explicit formatting instructions.

Experiences are constructed following Reasoning-Bank (Ouyang et al., 2025), which organizes both successful and failed model experiences into reusable experience entries. The experience-construction prompts are provided in Table 5 and Table 6.

For ReasoningBank, ExpeL, LLM-R, and IRCoT, they directly append retrieved experiences to the instruction during inference. We present illustrative prompts in Table 7, Table 8, Table 9, Table 11, and Table 12, corresponding to Question Answering, Math Reasoning, Coding, PharmaBench, and Recommendation tasks, respectively.

For Search-o1, it does not directly append retrieved experiences to the instruction; instead, it first summarizes them into a more helpful form. The experience-summarization prompt is provided in Table 13. We further report representative Search-o1 and Search-R1 prompt templates in Table 16, Table 14, Table 15, Table 17, and Table 18 for the same task categories.

For R1 and `ExpWeaver`, they do not explicitly incorporate retrieved experiences into the prompt during inference. Instead, `ExpWeaver` leverages a cross-attention mechanism to implicitly fuse experience representations into the model's hidden states. We present their prompt templates in Table 19, Table 20, Table 21, Table 22, and Table 23 for Question Answering, Math Reasoning, Coding, PharmaBench, and Recommendation tasks, respectively.

## B. Dataset Descriptions

We describe all 13 evaluation datasets below, categorized by their respective domains.

### B.1. Mathematical Reasoning

**GSM8K** (Cobbe et al., 2021a) comprises 8.5K elementary-level mathematical word problems that demand multi-step arithmetic calculations. Each problem necessitates 2-8 sequential reasoning operations involving fundamental arithmetic (addition, subtraction, multiplication, division). Evaluation employs exact matching on the final numeric solution.

**GSM-Symbolic** (Mirzadeh et al., 2025) represents a symbolic adaptation of GSM8K wherein numeric quantities are substituted with symbolic variables, thereby evaluating algebraic manipulation capabilities rather than pure numerical computation. This design enables assessment of genuine

*Table 5.* **Prompts for Summarizing Successful Trajectory into experience Item.**

```
<|im_start|>
system
You will be given a user query, the corresponding trajectory that represents how an agent successfully
accomplished the task.
Guidelines
You need to extract and summarize useful insights in the format of experience items based on the agent's
successful trajectory.
The goal of summarized experience items is to be helpful and generalizable for future similar tasks.
Important Notes
- You must first think why the trajectory is successful, and then summarize the insights.
- You can extract only 1 experience item from the trajectory.
- You must not repeat similar or overlapping items.
- Do not mention specific websites, queries, or string contents, but rather focus on the generalizable
insights.
Output Format
Your output must strictly follow the Markdown format shown below:
# experience Item 1
## Title: <the title of the experience item>
## Description: <one sentence summary of the experience item>
## Content: <1-3 sentences describing the insights learned to successfully accomplishing the task>
<|im_end|>
<|im_start|>
user
Query: {question}
Trajectory: {reasoning_trajectory}
<|im_end|>
```

mathematical comprehension versus pattern memorization.

**MATH** (Hendrycks et al., 2021c) encompasses 12.5K competition-level mathematics challenges across algebra, geometry, number theory, probability, counting, and pre-calculus. Sourced from prestigious competitions including AMC and AIME, these problems demand advanced multi-step reasoning combined with substantial domain expertise.

**NuminaMath** (out-of-domain) (Li et al., 2024) aggregates mathematical reasoning problems from international olympiads and competitions globally. This large-scale benchmark offers varied problem categories and complexity levels exceeding conventional datasets, thereby evaluating mathematical generalization under distribution shift.

### B.2. Question Answering

**MMLU** (Massive Multitask Language Understanding) (Hendrycks et al., 2021a) spans 57 academic subjects including STEM, humanities, and social sciences. Its multiple-choice format assesses both factual knowledge and reasoning capabilities across difficulty levels from elementary to professional expertise.

**CommonsenseQA** (Talmor et al., 2019a) presents 12.2K multiple-choice items demanding commonsense inference about quotidian concepts and relations. Derived from ConceptNet's knowledge graph, these questions necessitate understanding tacit world knowledge beyond explicit question

content.

**OpenBookQA** (Mihaylov et al., 2018a) features 5.9K elementary science queries following OpenBook examination formats. Each item requires synthesizing a provided scientific fact with supplementary commonsense knowledge, thereby testing multi-hop reasoning over scientific principles.

**ARC** (AI2 Reasoning Challenge) (Clark et al., 2018a) includes 7.8K natural science questions derived from standardized assessments. We utilize the Challenge subset (ARC-C), comprising questions resistant to simple retrieval and word co-occurrence heuristics, thus requiring authentic reasoning capabilities.

**GPQA** (Graduate-level Google-Proof QA) (Rein et al., 2023) constitutes a demanding benchmark of 448 multiple-choice problems in biology, physics, and chemistry. Designed to be search-resistant yet solvable by domain specialists, these questions evaluate deep disciplinary knowledge rather than surface-level information retrieval.

**SIQA** (Social Interaction QA, out-of-domain) (Sap et al., 2019) evaluates reasoning about human behaviors and their social consequences. Questions probe understanding of emotional responses, intentions, and interpersonal dynamics in commonplace scenarios, assessing social commonsense beyond factual knowledge.

*Table 6.* **Prompts for Summarizing Failed Trajectory into experience Item.**

```
<|im_start|>
system
You will be given a user query, the corresponding trajectory that represents how an agent attempted to
resolve the task but failed.
Guidelines
You need to extract and summarize useful insights in the format of experience items based on the agent's
failed trajectory.
The goal of summarized experience items is to be helpful and generalizable for future similar tasks.
Important Notes
- You must first reflect and think why the trajectory failed, and then summarize what lessons you have
learned or strategies to prevent the failure in the future.
- You can extract only 1 experience item from the trajectory.
- You must not repeat similar or overlapping items.
- Do not mention specific websites, queries, or string contents, but rather focus on the generalizable
insights.
Output Format
Your output must strictly follow the Markdown format shown below:
# experience Item 1
##Title: <the title of the experience item>
##Description: <one sentence summary of the experience item>
##Content: <1-3 sentences describing the insights learned to prevent similar failures in the future>
<|im_end|>
<|im_start|>
user
Query: {question}
Expected Answer: {expected_answer}
Trajectory: {reasoning_trajectory}
<|im_end|>
```

**PIQA** (Physical Interaction QA, out-of-domain) (Bisk et al., 2020) measures physical commonsense understanding regarding everyday objects and their interactions. Questions examine intuitive physics concepts including object functionalities, material characteristics, and physical causation typically acquired through embodied experience.

### B.3. Code Generation

**HumanEval+** extends the original HumanEval suite (Liu et al., 2023a) with augmented test coverage to minimize false positives. This benchmark comprises 164 programming challenges with provided function specifications and documentation strings, requiring generation of correct Python code satisfying all test constraints.

**MBPP+** augments the Mostly Basic Python Problems (Liu et al., 2023a) collection with enhanced test rigor. It encompasses 974 crowd-sourced Python programming tasks targeting entry-level competency, evaluating fundamental programming proficiency and standard algorithmic patterns.

### B.4. Chemistry

**TDC** (Therapeutics Data Commons) (Huang et al., 2021a) aggregates systematically organized data resources for drug development research, with particular emphasis on ADMET profiling—a collection of assays measuring Absorption, Dis-

tribution, Metabolism, Excretion, and Toxicity characteristics. Our experiments draw upon several ADMET classification problems where molecular structures encoded as SMILES strings serve as inputs, and binary labels indicate the presence or absence of specific pharmacological attributes. The prediction targets span diverse bioactivity profiles: intestinal absorption efficiency (HIA), central nervous system permeability (BBB), cytochrome P450 enzyme interactions (CYP2D6, CYP3A4), genetic mutation potential (Ames test), liver damage risk (hepatotoxicity), cardiac ion channel interference (hERG), and efflux transporter modulation (P-glycoprotein). Accurate prediction demands comprehension of three-dimensional molecular architecture, reactive chemical moieties, and pharmacokinetic principles governing drug disposition. Experimental validation through laboratory assays incurs substantial costs, thereby establishing this domain as an ideal testbed for annotation-efficient methods. The benchmark comprises approximately 150,000 compound-property annotations distributed across distinct assay types, facilitating thorough assessment of transfer learning to highly specialized scientific domains.

### B.5. Recommendation Ranking

**MovieLens ml-1m** (Hou et al., 2024) is a widely-used sequential recommendation dataset that comprises 1,000,209 explicit 1–5 ratings from 6,040 users on 4,000 movies, along

*Table 7.* **Prompts for Retrieval-Centric Baselines in Question Answering task**.

```
<|im_start|>
```
*system*
You are a helpful assistant skilled in logical reasoning.
Solve problems step by step, showing your work clearly.
CRITICAL FORMAT REQUIREMENT:
End your response with #### followed by just the final answer value.
```
<|im_end|>
<|im_start|>
```
*user*
Below are some experience items that I accumulated from past interaction from the environment that may be helpful to solve the task.
**{retrieved_experiences}**
Answer this question:
**{question}**
**{choices}**
Analyze each option and explain your reasoning.
Write your final answer as:
#### [letter]
```
<|im_end|>
```

with timestamps, user demographics, and movie genre metadata. It evaluates a model's ability to perform personalized sequential preference modeling and next-item ranking from relatively clean, moderately dense user histories.

**Amazon's CD and Vinyl** (McAuley et al., 2015; Ni et al., 2019b) is a classic sequential recommendation dataset that contains about 3.75M user–item interactions (star ratings/reviews) for CDs & Vinyl products along with accompanying product metadata. It is well-suited for evaluating sequential ranking methods under large-scale and noisy user behavior logs, thereby assessing robust sequential recommendation and long-tail generalization in realistic e-commerce settings.

## C. Dataset Statistics

In this section, we present detailed statistics for each dataset. Specifically, the statistics for ExpBench-Generic, ExpBench-Sci, and ExpBench-Rec are provided in Table 25, Table 26, and Table 27, respectively. For each dataset, we randomly sample 1,500 instances and divide them into training, validation, and test sets.

## D. Case studies

This appendix provides a comprehensive set of case studies illustrating the behavior of all methods across diverse scenarios. For each model, we report representative examples from five task categories: Question Answering, Math Reasoning, Coding, ParmaBench, and Recommendation.

Each case-study table contains:

- **Question**: the input query or prompt that initiates the task.

- **Ground Truth**: the reference answer.

- **Retrieved Experiences** (optional): the experience snippets retrieved and provided to the model.

- **Response**: the model-generated output.

These examples serve to highlight both the strengths and failure modes of each method, offering qualitative insights into their decision-making processes. ReasoningBank case studies are provided in Table 28 to Table 32, ExpeL case studies are provided in Table 33 to Table 37, LLM-R case studies are provided in Table 38 to Table 42, IRCoT case studies are provided in Table 43 to Table 47, Search-o1 case studies are provided in Table 48 to Table 48, R1 case studies are provided in Table 54 to Table 57, SearchR1 case studies are provided in Table 58 to Table 62, and ExpWeaver case studies are provided in Table 63 to Table 67.

*Table 8.* **Prompts for Retrieval-Centric Baselines in Math Reasoning task**.

```
<|im_start|>
```
*system*
You are a helpful assistant skilled in mathematics.
Solve problems step by step, showing your work clearly.
CRITICAL FORMAT REQUIREMENT:
End your response with #### followed by just the final answer value.
```
<|im_end|>
<|im_start|>
```
*user*
Below are some experience items that I accumulated from past interaction from the environment that may be helpful to solve the task.
**{retrieved_experiences}**
Solve this math word problem:
**{question}**
Think step by step, showing your calculations.
Write your final answer as:
#### [number]
```
<|im_end|>
```

*Table 9.* **Prompts for Retrieval-Centric Baselines in Coding task**.

```
<|im_start|>
```
*system*
You are an expert Python programmer.
Solve coding problems step by step.
CRITICAL FORMAT REQUIREMENT:
- You MUST end your response with #### followed by your complete code
- The code after #### should be in a ```python code block
- NEVER output code without the #### marker
```
<|im_end|>
<|im_start|>
```
*user*
Below are some experience items that I accumulated from past interaction from the environment that may be helpful to solve the task.
**{retrieved_experiences}**
Task:
**{task}**
Think through your approach, then write the Python function.
REMEMBER: You MUST format your response as:
####
```python
def solution():
    pass
```
```
<|im_end|>
```

*Table 10.* **Field mapping for the generated ADMET prompt in ParmaBench.**

| Prompt slot | Meaning / example |
| --- | --- |
| **{property_type}** | Key that specifies which ADMET task to predict (e.g., `ames`, `bbb`, `hia`). |
| **{property_name}** | Human-readable property name shown to the model (e.g., Ames Mutagenicity Test). |
| **{property_question}** | Binary classification question defining the task (e.g., "Is this molecule mutagenic according to Ames test?"). |
| **{positive_definition}** | Text definition of the label `Positive` (e.g., "Mutagenic (Ames+)"; "BBB permeable (BBB+)" ). |
| **{negative_definition}** | Text definition of the label `Negative` (e.g., "Non-mutagenic (Ames-)"; "BBB impermeable (BBB-)" ). |
| **{smiles}** | Input molecule in SMILES format inserted into the prompt. |
| **{mol_name}** (optional) | Optional molecule name prepended as "Molecule Name: {mol_name}". |

*Table 11.* **Prompts for Retrieval-Centric Baselines in TDC.**

```
<|im_start|>
```
*system*
You are an expert in computational chemistry and drug discovery.
Your task is to predict ADMET (Absorption, Distribution, Metabolism, Excretion, Toxicity)
properties of drug molecules based on their chemical structure.
CRITICAL FORMAT REQUIREMENT:
- First, write your step-by-step reasoning inside `<think></think>` tags
- Then, write your final answer inside `<answer></answer>` tags
- The answer must be exactly `Positive` or `Negative` for the asked property
Example format:
```
<think>
```
The molecule contains a nitro group which is a known toxicophore...
The molecular weight is within drug-like range...
The LogP suggests moderate lipophilicity...
```
</think>
<answer>Positive</answer>
<|im_end|>
<|im_start|>
```
*user*
Below are some experience items that I accumulated from past interaction from the environment that may be helpful to solve the task.
**{retrieved_experiences}**
Predict the **{property_name}** property of the following molecule.
**{mol_name}**
SMILES: **{smiles}**
Question: **{property_question}**
- Positive means: **{positive_definition}**
- Negative means: **{negative_definition}**

Analyze the molecular structure and functional groups, then predict.

Write your analysis inside `<think></think>` tags, then your answer inside `<answer></answer>` tags.
Answer must be exactly `Positive` or `Negative`.
```
<|im_end|>
```

*Table 12.* **Prompts for Retrieval-Centric Baselines in Recommendation task**.

---

`<|im_start|>`
*system*
You are a helpful assistant that ranks products by how likely the user is to buy them, based on their previous purchase history.
`<|im_end|>`
`<|im_start|>`
*user*
Below are some experience items that I accumulated from past interaction from the environment that may be helpful to solve the task.
**{retrieved_experiences}**
I've purchased the following items in the past, in order:
**{historical_interactions}**
Now there are 20 candidate items that I might purchase next:
**{candidate_items}**
Please rank these items by measuring the possibilities that I would like to buy next most, according to my purchase history. Please think step by step.
Split your output with line break. You MUST rank the given candidate items. You can not generate items that are not in the given candidate list. Show your work in `<think>` `<think>` tags. And return the final answer in `<answer>` `</answer>` tags.
`<|im_end|>`

---

*Table 13.* **Prompts for Summarizing Retrieved Experiences in Search-o1**

---

`<|im_start|>`
*user*
Task Instruction:

You are tasked with reading and analyzing experiences based on the following inputs: Previous Reasoning Steps, Current Search Query, and Searched Experiences. Your objective is to extract relevant and helpful information for the Current Search Query from the Searched Experiences and seamlessly integrate this information into the Previous Reasoning Steps to continue reasoning for the original question.
Guidelines:
1. Analyze the Searched Experiences:
- Carefully review the content of each searched experience.
- Identify factual information that is relevant to the Current Search Query and can aid in the reasoning process for the original question.
2. Extract Relevant Information:
- Select the information from the searched experiences that directly contributes to advancing the Previous Reasoning Steps.
- Ensure that the extracted information is accurate and relevant.

3. Output Format:
- If the experiences provide helpful information for the current search query: present the information beginning with `**Final Information**`.
`**Final Information**`
`[Helpful information]`
- If the experiences do not provide any helpful information: output the following text.
`**Final Information**`
No helpful information found.

Inputs:
- Previous Reasoning Steps: **{prev_reasoning}**
- Current Search Query: **{search_query}**
- Retrieved Experiences: **{retrieved_experiences}**
Now you should analyze each experience and find helpful information based on the current search query and previous reasoning steps.
`<|im_end|>`

---

*Table 14.* **Prompts for Search-o1 in Math**.

---

`<|im_start|>`

*system*

You are a helpful assistant skilled in mathematics. You have the ability to perform experience searches to help you answer the user's question accurately.

To perform a search: write `<|begin_search_query|>` your query here `<|end_search_query|>`.

Then, the system will search and analyze relevant experiences, then provide you with helpful information in the format `<|begin_search_result|>` ...search results... `<|end_search_result|>`.

Once you have all the information you need, continue your reasoning.

Solve problems step by step, showing your work clearly.

CRITICAL FORMAT REQUIREMENT:

- Use `<|begin_search_query|>` to request a experience search and end with `<|end_search_query|>`.

- When done searching, continue your reasoning.

- You MUST format your response as: #### [answer]

Example format:

Question: What is 7 + 5?

Assistant thinking steps:

- Add the two integers.

Assistant:

`<|begin_search_query|>`

7 + 5

`<|end_search_query|>`

`<|begin_search_result|>`

System returns processed information from relevant experiences

`<|end_search_result|>`

Assistant continues reasoning with the new information...

Assistant final output:

#### 12

`<|im_end|>`

`<|im_start|>`

*user*

`<|begin_search_query|>` **{summarized_experiences}** `<|end_search_query|>`

Solve this math word problem:

**{question}**

Think step by step, showing your calculations.

Write your final answer as:

#### [number]

`<|im_end|>`

---

*Table 15.* **Prompts for Search-o1 in Coding task**.

---

<|im_start|>
*system*

You are an expert Python programmer. You have the ability to perform experience searches to help you answer the user's question accurately.

To perform a search: write <|begin_search_query|> your query here <|end_search_query|>.

Then, the system will search and analyze relevant experiences, then provide you with helpful information in the format <|begin_search_result|> ...search results... <|end_search_result|>.

Once you have all the information you need, continue your reasoning.

Solve coding problems step by step.

CRITICAL FORMAT REQUIREMENT:

- You MUST end your response with #### followed by your complete code

- The code after #### should be in a ```python code block

- NEVER output code without the #### marker

Example format:

Task: Write a Python function to check if a string is a palindrome.

Assistant thinking steps:

- Normalize the string (lowercase, keep only letters/digits), then compare with its reverse.

Assistant:

<|begin_search_query|>

python remove non alphanumeric and check palindrome

<|end_search_query|>

<|begin_search_result|>

System returns processed information from relevant experiences

<|end_search_result|>

Assistant continues reasoning with the new information...

Assistant final output:

####

```python
def is_pal(s):
    pass
```

<|im_end|>

<|begin_search_query|> **{summarized_experiences}** <|end_search_query|>

<|end_search_query|>

Task:

**{task}**

Think through your approach, then write the Python function.

REMEMBER: You MUST format your response as:

####

```python
def solution():
    pass
```

<|im_end|>

---

*Table 16.* **Prompts for Search-o1 in QA task**.

---

`<|im_start|>`
*system*
You are a helpful assistant skilled in logical reasoning. You have the ability to perform experience searches to help you answer the user's question accurately.
To perform a search: write `<|begin_search_query|>` your query here `<|end_search_query|>`.
Then, the system will search and analyze relevant experiences, then provide you with helpful information in the format `<|begin_search_result|>` ...search results... `<|end_search_result|>`.
Once you have all the information you need, continue your reasoning.
Solve coding problems step by step.
CRITICAL FORMAT REQUIREMENT:
- You MUST end your response with #### followed by your complete code
- The code after #### should be in a ```python code block
- NEVER output code without the #### marker
Example format:
Question: Who got the first Nobel Prize in Physics?
Assistant thinking steps:
- I need to find out who was awarded the first Nobel Prize in Physics.
Assistant:
`<|begin_search_query|>`
first Nobel Prize in Physics winner
`<|end_search_query|>`
`<|begin_search_result|>`
System returns processed information from relevant experiences
`<|end_search_result|>`
Assistant continues reasoning with the new information...
Assistant final output:
#### Wilhelm Conrad Röntgen
`<|im_end|>`
`<|begin_search_query|>` **{summarized_experiences}** `<|end_search_query|>`
`<|end_search_query|>`
Task:
**{task}**
Think through your approach, then write the Python function.
REMEMBER: You MUST format your response as:
####
```python
def solution():
    pass
```
`<|im_end|>`

---

*Table 17.* **Prompts for Search-o1 in TDC.**

---

```
<|im_start|>
```
*system*

You are an expert in computational chemistry and drug discovery.

You have the ability to perform experience searches to help you answer the user's question accurately.

To perform a search: write `<|begin_search_query|>` your query here `<|end_search_query|>`.

Then, the system will search and analyze relevant experiences, then provide you with helpful information in the format `<|begin_search_result|>` ...search results... `<|end_search_result|>`.

Once you have all the information you need, continue your reasoning.

Your task is to predict ADMET (Absorption, Distribution, Metabolism, Excretion, Toxicity) properties of drug molecules based on their chemical structure.

CRITICAL FORMAT REQUIREMENT:

- First, write your step-by-step reasoning inside `<think></think>` tags

- Then, write your final answer inside `<answer></answer>` tags

- The answer must be exactly `Positive` or `Negative` for the asked property

Example format:

Question: Based on the molecule's chemical structure (SMILES: O=[N+]([O-])c1ccc(N)cc1), predict whether it has DILI (drug-induced liver injury) risk.

Assistant thinking steps:

- Identify key functional groups and structural alerts in the SMILES (e.g., nitro-aromatic, aniline).

Assistant:

```
<|begin_search_query|>
```
ADMET structural alerts nitro aromatic aniline reactive metabolites DILI
```
<|end_search_query|>
```
```
<|begin_search_result|>
```
System returns processed information from relevant experiences
```
<|end_search_result|>
```
Assistant continues reasoning with the new information...

Assistant final output:
```
<think>
```
The molecule contains a nitro group which is a known toxicophore...

The molecular weight is within drug-like range...

The LogP suggests moderate lipophilicity...
```
</think>
```
`<answer>`Positive`</answer>`
```
<|im_end|>
```
```
<|im_start|>
```
*user*

`<|begin_search_query|>` **{summarized_experiences}** `<|end_search_query|>`

Predict the **{property_name}** property of the following molecule. **{mol_name}**

SMILES: **{smiles}**

Question: **{property_question}**

- Positive means: **{positive_definition}**

- Negative means: **{negative_definition}**

Analyze the molecular structure and functional groups, then predict.

Write your analysis inside `<think></think>` tags, then your answer inside `<answer></answer>` tags.

Answer must be exactly `Positive` or `Negative`.
```
<|im_end|>
```

---

*Table 18.* **Prompts for Search-o1 in Recommendation task**.

```
<|im_start|>
```
*system*

You are a helpful assistant that ranks products by how likely the user is to buy them, based on their previous purchase history.

You have the ability to perform experience searches to help you answer the user's question accurately.

To perform a search: write `<|begin_search_query|>` your query here `<|end_search_query|>`.

Then, the system will search and analyze relevant experiences, then provide you with helpful information in the format `<|begin_search_result|>` ...search results... `<|end_search_result|>`.

Once you have all the information you need, continue your reasoning.

Example format:

Question: I have purchased the following items in the past, in order: ... Now these are candidate items that I might purchase next: ... Please rank these items.

Assistant thinking steps:

- Infer user preferences from purchase history (category/brand/theme).

- Score each candidate by similarity and complementarity, then sort.

Assistant:

```
<|begin_search_query|>
```
science fiction
```
<|end_search_query|>
<|begin_search_result|>
```
next-item ranking based on purchase history similarity complementarity
```
<|end_search_result|>
```
Assistant continues reasoning with the new information...

Assistant final output:

```
<think>
```
Candidates similar to recent purchases are ranked higher; complementary items are promoted.
```
</think>
<answer>
```
{item_1}

{item_2}

...

{item_20}

```
</answer>
<|im_end|>
<|im_start|>
```
*user*
```
<|begin_search_query|>
```
**{summarized_experiences}**
```
<|end_search_query|>
```
I've purchased the following items in the past, in order:

**{historical_interactions}**

Now there are 20 candidate items that I might purchase next:

**{candidate_items}**

Please rank these items by measuring the possibilities that I would like to buy next most, according to my purchase history. Please think step by step.

Split your output with line break. You MUST rank the given candidate items. You can not generate items that are not in the given candidate list. Show your work in `<think>` `<think>` tags. And return the final answer in `<answer>` `</answer>` tags.
```
<|im_end|>
```

*Table 19.* **Prompts for R1 and `ExpWeaver` in Question Answering task**.

---

```
<|im_start|>
```
*system*
You are a helpful assistant skilled in logical reasoning.
Solve problems step by step, showing your work clearly.
CRITICAL FORMAT REQUIREMENT:
- You MUST end your response with #### followed by your final answer
- NEVER use \boxed{}, $\boxed{}$, or any LaTeX boxing format
- ALWAYS use #### format, even for mathematical expressions
```
<|im_end|>
```
```
<|im_start|>
```
*user*
Answer this question:
**{question}**
**{choices}**
Analyze each option and explain your reasoning.
Write your final answer as:
#### [letter]
```
<|im_end|>
```

---

*Table 20.* **Prompts for R1 and `ExpWeaver` in Math Reasoning task**.

---

```
<|im_start|>
```
*system*
You are a helpful assistant skilled in mathematics.
Solve problems step by step, showing your work clearly.
CRITICAL FORMAT REQUIREMENT:
- You MUST end your response with #### followed by your final answer
- NEVER use \boxed{}, $\boxed{}$, or any LaTeX boxing format
- ALWAYS use #### format, even for mathematical expressions
```
<|im_end|>
```
```
<|im_start|>
```
*user*
Solve this math word problem:
**{question}**
Think step by step, showing your calculations.
Write your final answer as:
#### [number]
```
<|im_end|>
```

---

*Table 21.* **Prompts for R1 and `ExpWeaver` in Coding task**.

```
<|im_start|>
```
*system*
You are an expert Python programmer.
Solve coding problems step by step.
CRITICAL FORMAT REQUIREMENT:
- You MUST end your response with #### followed by your complete code
- The code after #### should be in a ```python code block
- NEVER output code without the #### marker
```
<|im_end|>
<|im_start|>
```
*user*
Task: **{task_description}**
Function signature: **{function_signature}**
Your code should pass these tests:
**{test_cases}**
Think through your approach, then write the Python function.
REMEMBER: You MUST format your response as:
####
```python
def solution():
    pass
```
```
<|im_end|>
```

*Table 22.* **Prompts for R1 and** `ExpWeaver` **in TDC**.

---

`<|im_start|>`
*system*
You are an expert in computational chemistry and drug discovery.
Your task is to predict ADMET (Absorption, Distribution, Metabolism, Excretion, Toxicity)
properties of drug molecules based on their chemical structure.
CRITICAL FORMAT REQUIREMENT:
- First, write your step-by-step reasoning inside `<think></think>` tags
- Then, write your final answer inside `<answer></answer>` tags
- The answer must be exactly `Positive` or `Negative` for the asked property
Consider these factors in your analysis:
- Functional groups (hydroxyl, amine, carboxyl, etc.)
- Molecular weight and lipophilicity (LogP)
- Structural alerts for toxicity (nitro groups, reactive carbonyls, etc.)
- Ring systems and aromatic character
- Hydrogen bond donors and acceptors
Example format:
`<think>`
The molecule contains a nitro group which is a known toxicophore...
`</think>`
`<answer>`Positive`</answer>`
`<|im_end|>`
`<|im_start|>`
*user*
Predict the **{property_name}** property of the following molecule.
**{mol_name}**
SMILES: **{smiles}**
Question: **{property_question}**
- Positive means: **{positive_definition}**
- Negative means: **{negative_definition}**

Analyze the molecular structure and functional groups, then predict.

Write your analysis inside `<think></think>` tags, then your answer inside
`<answer></answer>` tags.
Answer must be exactly `Positive` or `Negative`.
`<|im_end|>`

---

*Table 23.* **Prompts for R1 and `ExpWeaver` in Recommendation task.**

```
<|im_start|>
```
*system*
You are a helpful assistant that ranks products by how likely the user is to buy them, based on their previous purchase history.
CRITICAL FORMAT REQUIREMENT:
- Show your reasoning process inside `<think></think>` tags
- Then, provide your final ranking inside `<answer></answer>` tags
```
<|im_end|>
<|im_start|>
```
*user*
I've purchased the following items in the past, in order:
**{historical_interactions}**
Now there are 20 candidate items that I might purchase next:
**{candidate_items}**
Please rank these items by measuring the possibilities that I would like to buy next most, according to my purchase history. Please think step by step.
Split your output with line break. You MUST rank the given candidate items. You can not generate items that are not in the given candidate list.
Show your work in `<think></think>` tags. And return the final answer in `<answer></answer>` tags.
```
<|im_end|>
```

*Table 24.* **Detailed summarization of tasks used in our agent experience learning experiments.** We categorize datasets by scenario, task type, and evaluation metric.

| Dataset | Task | Metric |
|---|---|---|
| **ExpBench-Generic** | | |
| ARC-C | Question Answering | Accuracy |
| CommonsenseQA | Question Answering | Accuracy |
| GPQA | Question Answering | Accuracy |
| MMLU | Question Answering | Accuracy |
| OBQA | Question Answering | Accuracy |
| GSM8K | Reasoning | Exact Match |
| GSM-Symbolic | Reasoning | Exact Match |
| MATH | Reasoning | Exact Match |
| HumanEval+ | Coding | Pass@1 |
| MBPP+ | Coding | Pass@1 |
| **ExpBench-Sci** | | |
| Chem-TDC | Pharmaceutical | Exact Match |
| **ExpBench-Rec** | | |
| Rec-Movie | Ranking | NDCG@10 / MRR |
| Rec-Music | Ranking | NDCG@10 / MRR |

*Table 25.* **ExpBench-Generic Data Statistics.**

| Split | Math | | | QA | | | | | Code | |
|---|---|---|---|---|---|---|---|---|---|---|
| | GSM8K | GSM-sym | MATH | MMLU | CSQA | OBQA | ARC-C | GPQA | HumanEval+ | MBPP+ |
| Train | 750 | 750 | 750 | 750 | 750 | 750 | 750 | 99 | 65 | 132 |
| Valid | 300 | 300 | 300 | 300 | 300 | 300 | 300 | 39 | 26 | 52 |
| Test | 450 | 450 | 450 | 450 | 450 | 450 | 450 | 60 | 39 | 80 |

*Table 26.* **ExpBench-Sci Data Statistics.**

| Split | PharmaBench |
|---|---|
| Train | 750 |
| Valid | 300 |
| Test | 450 |

*Table 27.* **ExpBench-Rec Data Statistics**.

| Split | Rec-Movie | Rec-Music |
|---|---|---|
| Train | 329 | 329 |
| Valid | 131 | 131 |
| Test | 197 | 197 |

*Table 28.* **ReasoningBank's case study in QA.**

**Question**: How does a tiger get stripes?
A. from its environment
B. from its food
C. from its offspring
D. from its parents

**Ground Truth**: D

**Retrieved experience 1 (✓Success Strategy)**:
**Inherited Trait for Polar Bear Fur Color**
Polar bears have white fur, which is an inherited trait. This fur color is not due to external snow but is passed down genetically. The analysis of the given options confirms that the correct answer is that the white fur is an inherited characteristic.
**Retrieved experience 2 (✓Success Strategy)**:
**Inherited Traits**
Inherited traits are not fixed and can vary significantly between individuals within the same family. Traits such as long hair, long eyelashes, and small dimples are not definitive inherited traits and can be influenced by environmental factors.
**Retrieved experience 3 (✗Lesson from Failure)**:
**Parental Instinct Transmission**
Bears have a strong genetic component to their instincts, and parents play a crucial role in passing on survival and behavioral instincts, including the instinct to hibernate.

A is incorrect because stripes are not caused by environment. B is incorrect because diet does not determine coat patterns. C is incorrect because offspring inherit traits but do not cause them. D is correct: stripes are a genetic trait inherited from parents.
#### D

*Table 29.* **ReasoningBank's case study in Math.**

| |
|---|
| **Question**: One-half of a pound of mangoes costs $0.60. How many pounds can Kelly buy with $12? |
| **Ground Truth**: 10 |

**Retrieved experience 1 (✗Lesson from Failure)**:
**Division to Find Unit Cost**
To accurately determine the unit cost of each lemon, it is crucial to perform the division correctly, ensuring that the total cost is divided by the number of lemons in the bag.

**Retrieved experience 2 (✗Lesson from Failure)**:
**Eraser Price Calculation**
Recognizing the proportional relationship between the prices of pencils and erasers is essential for accurately calculating the cost of erasers in the given scenario.

**Retrieved experience 3 (✗Lesson from Failure)**:
**Cost Calculation and Division for Determining Quantity**
The agent correctly calculated the cost per can of pears and used division to find out how many cans can be bought for a specific amount of money. However, the agent did not round down the result to the nearest whole number, which led to an incorrect final answer. To prevent this failure, it is crucial to ensure that the final answer is a whole number, as partial cans are not typically sold.

To solve this problem, we'll follow these steps:
1. Determine the cost per pound of mangoes:
   - We know that one-half of a pound costs $0.60.
   - To find the cost of one pound, we multiply the cost of one-half pound by 2.
   - Calculation: $0.60 × 2 = $1.20
2. Calculate how many pounds Kelly can buy with $12:
   - We have the cost per pound ($1.20) and the total amount Kelly has ($12).
   - To find the number of pounds, we divide the total amount by the cost per pound.
   - Calculation: $12 ÷ $1.20
Let's perform the division:
   - $12 ÷ $1.20 = 10
Therefore, Kelly can buy 10 pounds of mangoes with $12.
#### 10

*Table 30.* **ReasoningBank's case study in Coding.**

**Question**: Task: Given an array `arr` of integers, find the minimum number of elements that need to be changed to make the array palindromic. A palindromic array is an array that is read the same backwards and forwards. In one change, you can change one element to any other element.
Examples:
```
smallest_change([1,2,3,5,4,7,9,6]) == 4
smallest_change([1,2,3,4,3,2,2]) == 1
smallest_change([1,2,3,2,1]) == 0
```
Tests:
```
assert smallest_change([1,2,3,5,4,7,9,6]) == 4
assert smallest_change([1,2,3,4,3,2,2]) == 1
assert smallest_change([1,4,2]) == 1
```

**Ground Truth**:
```
def smallest_change(arr):
  ans = 0
  for i in range(len(arr) // 2):
    if arr[i] != arr[len(arr) - i - 1]:
      ans += 1
  return ans
```

**Retrieved experience 1 (✓Success Strategy)**:
**Incrementing and Checking Palindromes**
To find the next smallest palindrome, the agent increments the given number and checks if the resulting number is a palindrome. If it is, the agent returns the number as the next smallest palindrome. If not, the agent continues incrementing and checking until a palindrome is found.
**Retrieved experience 2 (✓Success Strategy)**:
**Palindrome Check Logic**
The function iterates over the string, comparing the first and last characters. If they are not the same, the string is not a palindrome. If the string is empty or has one character, it is considered a palindrome. The function continues to check the substring excluding the first and last characters until the entire string is checked or a mismatch is found.
**Retrieved experience 3 (✓Success Strategy)**:
**Remove Characters and Check Palindrome**
To solve the task, we first remove all characters from the string `s` that are present in the string `c` using a list comprehension. Then, we check if the resulting string is a palindrome by comparing it to its reverse. Finally, we return a tuple containing the modified string and a boolean indicating whether it is a palindrome.

####
```python
def smallest_change(arr):
  def is_palindrome(arr):
    return arr == arr[::-1]

  left, right = 0, len(arr) - 1
  changes = 0

  while left < right:
    if arr[left] != arr[right]:
      changes += 1
    left += 1
    right -= 1

  return changes
```

*Table 31.* **ExpeL's case study in TDC.**

---

**Question**: Predict the Ames Mutagenicity Test property of the following molecule.
Molecule Name: Drug 2725
SMILES: `Nc1ccc2nc3c(N)cccc3nc2c1`
Question: Is this molecule mutagenic according to Ames test?
- Positive means: Mutagenic (Ames+)
- Negative means: Non-mutagenic (Ames-)

---

**Ground Truth**: Positive

---

**Retrieved experience 1 (✓Success Strategy)**:
**Key Functional Group for Ames Test Mutagenicity**
The carbonyl group (C#N) and carboxyl group (CO) in the molecular structure of Drug 296 are key functional groups that suggest the molecule may be mutagenic, as these groups can form reactive intermediates and are known to be mutagenic in the Ames test.
**Retrieved experience 2 (✓Success Strategy)**:
**Functional Groups and Structural Features as Indicators of Ames Mutagenicity**
The molecule contains a carboxyl group, an amine, and a carbonyl group, which are known to be mutagenic. However, the presence of these groups alone is not definitive. The specific arrangement and the presence of other structural features are crucial, and additional data such as molecular weight, LogP, and specific structural alerts are needed for a definitive judgment.
**Retrieved experience 3 (✓Success Strategy)**:
**Key Functional Groups and Structural Features Predicting Ames Mutagenicity**
The molecule contains a nitro group and a carbonyl group, both of which are known to be mutagenic. Additionally, the presence of a benzene ring can lead to reactive intermediates, increasing the likelihood of DNA damage and thus mutagenicity.

---

`<think>` To predict the Ames Mutagenicity Test property of the molecule, we need to consider the presence of certain functional groups and structural features that are known to be mutagenic. The given SMILES string represents a molecule with a nitrogen atom, a carbonyl group, and a pyrrole ring.
1. **Nitrogen Atom**: The presence of a nitrogen atom can be a potential source of reactive species, which might be mutagenic. However, the specific configuration and the presence of other groups around it are crucial.
2. **Carbonyl Group**: The carbonyl group (C=O) can also be a source of reactive intermediates. However, the presence of a pyrrole ring can stabilize these reactive intermediates and reduce the likelihood of mutagenicity.
3. **Pyrrole Ring**: The pyrrole ring (N-C-C-C-N) is a common structural feature in many natural products and can be less likely to be mutagenic compared to other aromatic rings. However, the presence of a nitrogen atom in the ring can still be a concern.
Given the presence of a nitrogen atom and a carbonyl group, the molecule has some potential for generating reactive species. However, the presence of the pyrrole ring might mitigate some of these risks.
Considering the above points, the molecule is likely to be mutagenic. `</think>`
`<answer>` Positive `</answer>`

*Table 32.* **ReasoningBank's case study in Movie Ranking Scenario.**

**Question**: Please rank the following movies based on the likelihood that I would like to watch them next, according to my previous viewing history.
Movies Watched: ['0. Cool Hand Luke', '1. Rain Man', '2. Night of the Living Dead', '3. North by Northwest', '4. Little Big Man', '5. Butch Cassidy and the Sundance Kid', '6. Dances with Wolves', '7. High Noon', '8. Fistful of Dollars, A', '9. Blazing Saddles', '10. Hud', '11. Angel and the Badman', '12. Magnum Force', '13. Outlaw Josey Wales, The', '14. Unforgiven', '15. Jeremiah Johnson', '16. Good, The Bad and The Ugly, The', '17. High Plains Drifter', '18. Pale Rider', '19. Legends of the Fall']
Candidate Movies: ['0. Fried Green Tomatoes', '1. Hang 'em High', '2. Kazaam', '3. Kiss the Girls', '4. Little Rascals, The', '5. Space Cowboys', '6. Unstrung Heroes', '7. Golden Earrings', '8. Roadside Prophets', '9. What Lies Beneath', '10. Where the Money Is', '11. Blue Hawaii', '12. Heat', '13. Bloodsport 2', '14. Thelma & Louise', '15. Big Trouble in Little China', '16. Messenger: The Story of Joan of Arc, The', '17. Frances', '18. Bad Lieutenant', '19. Fall']

**Ground Truth**: Hang 'em High

**Retrieved experience 1 (✗Lesson from Failure)**:
\*\*Ranking of Candidate Movies\*\*
Based on the user's past preferences, the most likely next movies to watch are 'Tombstone', 'Being John Malkovich', and 'Pulp Fiction'. These movies are either in the user's watched list or are in the same genre, increasing the likelihood of the user choosing them.

**Retrieved experience 2 (✗Lesson from Failure)**:
\*\*Ranking of Candidate Movies\*\*
Based on the user's watching history, the most likely movies to be watched next are those that share similar genres and themes. The ranking of the candidate movies is as follows: 1. Beauty and the Beast, 2. October Sky, 3. Searching for Bobby Fischer, 4. Empire Records, 5. James and the Giant Peach, 6. Anastasia, 7. Pocahontas, 8. The Secret Garden, 9. Quiz Show, 10. Mad Love, 11. Sudden Manhattan, 12. 52 Pick-Up, 13. Boys of St. Vincent, The, 14. Midnight Run, 15. Benny & Joon, 16. Red Dawn, 17. Welcome To Sarajevo, 18. What Ever Happened to Baby Jane?, 19. Ring, The, 20. Exorcist II: The Heretic.

**Retrieved experience 3 (✗Lesson from Failure)**:
\*\*Ranking of Candidate Movies\*\*
After matching the genres and themes, the most likely movies to be watched next are 'High Plains Drifter', 'Days of Heaven', and 'Fried Green Tomatoes'.

`<think>` To rank the given candidate movies based on the user's past movie preferences, we need to analyze the user's preferences from their past movie list. Here's the step-by-step reasoning:
1. \*\*Identify Common Genres and Themes\*\*:
- The user has a diverse range of genres in their viewing history, including drama (e.g., 'Pulp Fiction', 'American Beauty'), thriller (e.g., 'Run Lola Run', 'Henry: Portrait of a Serial Killer'), comedy (e.g., 'His Girl Friday', 'Chicken Run'), and horror (e.g., 'Rosemary's Baby').
- This indicates that the user enjoys various genres, which will influence the ranking of candidate movies.
2. \*\*Ranking the Candidate Movies\*\*:
- Based on the user's interest in Westerns and Drama films, we will prioritize those genres. The candidate movies related to Westerns, Drama, and Action/Adventure will be ranked higher.
3. \*\*Final Ranking\*\*:
- \*\*1. Hang 'em High\*\* (Western - High similarity with the user's interest in Westerns like 'High Noon' and 'Unforgiven')
- \*\*2. Fried Green Tomatoes\*\* (Drama - Aligns with the user's interest in dramatic films like 'American Beauty')
- \*\*3. Unstrung Heroes\*\* (Drama - Matches with the user's preference for drama films like 'American Beauty')
- \*\*4. Space Cowboys\*\* (Action/Drama - Similar to 'Dances with Wolves' and other action/dramas)
- \*\*5. Golden Earrings\*\* (Drama - Matches with dramatic films like 'The Beach')
... `</think>`
`<answer>` Fried Green Tomatoes, Hang 'em High, Unstrung Heroes, Space Cowboys, Golden Earrings, What Lies Beneath, Where the Money Is, Blue Hawaii, Heat, Bloodsport 2, Thelma & Louise, Big Trouble in Little China, Messenger: The Story of Joan of Arc, The, Frances, Bad Lieutenant, Fall `</answer>`

*Table 33.* **ExpeL's case study in QA.**

---

**Question**: Although they belong to the same family, an eagle and a pelican are different. What is one difference between them?
A. their preference for eating fish
B. their ability to fly
C. their method of reproduction
D. their method of catching food

---

**Ground Truth**: D

---

**Retrieved experience 1 (✓ Success Strategy)**:
**Pelican's Post-Eating Behavior**
After eating a dart frog, a pelican is most likely to fly home as a plausible and logical continuation of the scenario, given the context and typical bird behavior.
**Retrieved experience 2 (✓ Success Strategy)**:
**Visual Communication in Birds**
Bright, multi-colored feathers in birds are most likely used to attract mates or prey, rather than to find the mother of future children, attract a plane to rescue them, or become famous in movies. This insight is based on the biological function of plumage in avian species.
**Retrieved experience 3 (✓ Success Strategy)**:
**Webbed Feet as a Unique Trait**
Having webbed feet is a unique trait that applies only to fish, as it is not present in reptiles, mammals, birds, or amphibians.

---

Option A is not a clear differentiator because both can eat fish. Option B is incorrect since both can fly. Option C is incorrect because both lay eggs. Option D is correct: eagles typically hunt by swooping/diving to seize prey, while pelicans characteristically scoop fish with a large bill/pouch.
#### D

---

*Table 34.* **ExpeL's case study in Math.**

---

**Question**: Trevor is a comic book illustrator. In the past three months, he has drawn 220 pages of the comic. The comic releases one issue per month. The first and second months' issues were the same number of pages, but the third month's issue was four pages longer than either of the others. How many pages was the first issue?

---

**Ground Truth**: 72

---

**Retrieved experience 1 (✓ Success Strategy)**:
**Reading Rate Calculation**
Jerry's reading rate is 20 pages per hour, which can be calculated by dividing the number of pages he reads by the time it takes. This rate is consistent and can be used to predict the time needed to read any number of pages.
**Retrieved experience 2 (✓ Success Strategy)**:
**Calculating Remaining Muffins and Required Boxes**
To find the number of boxes needed, the number of muffins left after some are sent to a hotel should be divided by the capacity of each box. This insight is useful for similar problems involving distribution of items into containers of a fixed size.
**Retrieved experience 3 (✓ Success Strategy)**:
**Subtraction for Comparing Mileage**
To compare the number of miles driven in different months, the agent subtracted the number of miles driven in May from the number of miles driven in April. This approach is useful for determining the difference in quantities between two time periods.

---

To solve this problem, let's denote the number of pages in the first and second months' issues as $x$. According to the problem, the third month's issue had 4 pages more than either of the first two months' issues, so it had $x + 4$ pages.
The total number of pages drawn in the three months is given as 220 pages. Therefore, we can set up the following equation to represent the total pages drawn: $x + x + (x + 4) = 220$
Simplify the equation: $3x + 4 = 220$
Subtract 4 from both sides to isolate the term with $x$: $3x = 216$
Divide both sides by 3 to solve for $x$: $x = 72$
Thus, the number of pages in the first issue of the comic is:
#### 72

*Table 35.* **ExpeL's case study in Coding.**

**Question**: Task: Return the largest prime factor of $n$. Assume $n > 1$ and is not a prime.
```
>>> largest_prime_factor(13195) → 29
>>> largest_prime_factor(2048) → 2
```
Tests:
```
assert largest_prime_factor(15) == 5
assert largest_prime_factor(27) == 3
assert largest_prime_factor(63) == 7
```

**Ground Truth**:
```python
def largest_prime_factor(n:  int):
  def is_prime(k):
    if k < 2:
      return False
    for i in range(2, k - 1):
      if k % i == 0:
        return False
    return True
  largest = 1
  for j in range(2, n + 1):
    if n % j == 0 and is_prime(j):
      largest = max(largest, j)
  return largest
```

**Retrieved experience 1 (✓Success Strategy)**:
**Prime Number Check**
To solve the problem, we need to check if the given number `n` is a prime number. If `n` is prime, we return `x`; otherwise, we return `y`.

**Retrieved experience 2 (✓Success Strategy)**:
**Prime Check Function**
The function `is_prime_hex_digit` checks if a given hexadecimal digit (2, 3, 5, 7, B, D) is a prime number.

**Retrieved experience 3 (✓Success Strategy)**:
**Prime Number Check**
To determine if a number is prime, it is sufficient to check divisibility up to the square root of the number, as any factors larger than the square root would have a corresponding factor smaller than the square root.

```
####
```python import math
def largest_prime_factor(n:  int):
  # Initialize the largest prime factor
  largest_prime = -1

  # Divide n by 2 to remove all even factors
  while n % 2 == 0:
    largest_prime = 2
    n //= 2

  # n must be odd at this point, so we can skip one element (Note i = i
+ 2)
  for i in range(3, int(math.sqrt(n)) + 1, 2):
    # While i divides n, update n and largest_prime
    while n % i == 0:
      largest_prime = i
      n //= i

  # This condition is to check if n is a prime number greater than 2
  if n > 2:
    largest_prime = n

  return largest_prime
```
```

*Table 36.* **ReasoningBank's case study in TDC.**

---

**Question**: Predict the Ames Mutagenicity Test property of the following molecule.
Molecule Name: Drug 3111
SMILES: `O=C(Cl)c1cccc(Cl)c1`
Question: Is this molecule mutagenic according to Ames test?
- Positive means: Mutagenic (Ames+)
- Negative means: Non-mutagenic (Ames-)

---

**Ground Truth**: Positive

---

**Retrieved experience 1 (✓Success Strategy)**:
**Ames Mutagenicity Prediction Insights**
The presence of specific functional groups like carboxyl and nitro groups can be indicative of mutagenicity, but the absence of such alerts and the typical molecular characteristics of drug-like molecules do not definitively predict Ames mutagenicity.
**Retrieved experience 2 (✓Success Strategy)**:
**Chlorine and Carbonyl Group as Key Features for Ames Test Prediction**
The chlorine atom and the carbonyl group are common features in mutagenic compounds. The chlorine atom is known to be a mutagenic and toxicophore, and the carbonyl group can form reactive intermediates that might be involved in mutagenesis. These features, combined with the absence of clear ring systems or aromatic character, indicate that the molecule is likely to be mutagenic.
**Retrieved experience 3 (✓Success Strategy)**:
**Functional Group Analysis for Ames Test Prediction**
The presence of a fluorine atom in the molecule does not significantly alter its mutagenicity profile. Functional groups like nitro groups, alkenes, and reactive functional groups are typically associated with mutagenicity, but the fluorine atom in this case does not contribute to mutagenicity.

---

`<think>` The given SMILES string represents the molecule 3-(chloroethyl)-1H-indole-2-carbaldehyde. Let's analyze its structure:
1. **Functional Groups**: The molecule contains a chloroethyl group (-ClCH2) and an indole ring. The presence of a chlorine atom is a key factor in determining mutagenicity. Chlorine is known to be a strong electrophile and can react with DNA, potentially leading to mutations.
2. **Indole Ring**: The indole ring is a common structural feature in many known mutagens. It can form covalent bonds with DNA bases, leading to mutations.
3. **Molecular Structure**: The structure is relatively simple, but the presence of a chlorine atom in a reactive position (as part of a functional group) is a significant indicator of potential mutagenicity.
Given the presence of a chlorine atom in a reactive position and the presence of an indole ring, which is known to be mutagenic, it is highly likely that this molecule would be mutagenic in the Ames test.
`</think>`
`<answer>` Positive `</answer>`

*Table 37.* **ExpeL's case study in Movie Ranking Scenario.**

**Question**: Please rank the following movies based on the likelihood that I would like to watch them next, according to my previous viewing history.
Movies Watched: ['0. M*A*S*H', '1. Titanic', '2. Dirty Dancing', '3. Rain Man', '4. Backdraft', '5. Strangers on a Train', '6. What Happened Was...', '7. Dr. Strangelove or: How I Learned to Stop Worrying and Love the Bomb', '8. Inherit the Wind', '9. Blade Runner', '10. Godfather, The', '11. Shawshank Redemption, The', '12. Seven Samurai', '13. Citizen Kane', '14. Wizard of Oz, The', '15. Battleship Potemkin, The', '16. American Beauty', '17. Grapes of Wrath, The', '18. Taxi Driver', '19. Raging Bull']
Candidate Movies: ['0. Sunset Blvd.', '1. Farewell to Arms, A', '2. BASEketball', '3. Spice World', '4. Country', '5. Children of Paradise', '6. Dark Half, The', '7. Nineteen Eighty-Four', '8. Kolya', '9. Night of the Living Dead', '10. Coogan's Bluff', '11. Rosetta', '12. Easy Rider', '13. Cool Dry Place, A', '14. Freeway', '15. My Man Godfrey', '16. Heathers', '17. What Ever Happened to Baby Jane?', '18. Up in Smoke', '19. Eden']

**Ground Truth**: Children of Paradise

**Retrieved experience 1 (✓Success Strategy)**:
**Genre Similarity and Personal Preference**
The agent identified that movies with similar genres or themes to the movies watched in the past are more likely to be chosen. This includes comedies, dramas, and action films. Therefore, when recommending movies, it is important to consider the genre and themes of the movies the user has previously enjoyed.
**Retrieved experience 2 (✓Success Strategy)**:
**Movie Similarity Analysis**
The agent analyzed the themes and genres of the movies watched in the past and found that movies with similar themes and genres are more likely to be watched next. This analysis helps in ranking the candidate movies based on their similarity to the past viewing history.
**Retrieved experience 3 (✓Success Strategy)**:
**Movie Similarity Heuristic**
The agent used a heuristic method to rank the candidate movies by comparing them to the movies in the user's watching history. Movies that were similar to the user's past preferences were given higher likelihood of being liked.

`<think>` irst, let's analyze the genres and themes of the movies in the user's past watching history:
1. **0. M*A*S*H** - Comedy, Drama
2. **1. Titanic** - Drama, Romance
3. **2. Dirty Dancing** - Comedy, Drama
4. **3. Rain Man** - Drama, Comedy
5. **4. Backdraft** - Drama, Action
...
Now, let's analyze the candidate movies:
1. **0. Sunset Blvd.** - Drama, Romance
2. **1. Farewell to Arms, A** - Drama, Romance
3. **2. BASEketball** - Sports, Comedy
4. **3. Spice World** - Comedy, Romance
5. **4. Country** - Comedy, Drama
...
Now, let's rank the candidate movies based on their similarity to the user's past preferences:
1. **0. Sunset Blvd.** - Drama, Romance (High similarity to 'Titanic', 'Rain Man', 'Wizard of Oz')
2. **1. Farewell to Arms, A** - Drama, Romance (High similarity to 'Titanic', 'Rain Man', 'Wizard of Oz')
3. **5. Children of Paradise** - Drama, Romance (High similarity to 'Titanic', 'Rain Man', 'Wizard of Oz')
4. **11. Coogan's Bluff** - Drama, Action (High similarity to 'Backdraft', 'Seven Samurai')
5. **12. Rosetta** - Drama, History (High similarity to 'Grapes of Wrath, The', 'Citizen Kane')
... `</think>`

`<answer>` 0. Sunset Blvd., 1. Farewell to Arms, A, 5. Children of Paradise, 11. Coogan's Bluff, 12. Rosetta `</answer>`

*Table 38.* **LLM-R's case study in QA.**

---

**Question**: To determine the most complete list of forms of tobacco that cause cancer, choose the best option.
Options:
A. cigarettes and snuff
B. chewing tobacco and snuff
C. cigarettes and chewing tobacco
D. cigarettes, chewing tobacco, and snuff

---

**Ground Truth**: D

---

**Retrieved experience 1 (✗Lesson from Failure)**:
**Relatively Prime Condition and Factor Pairs**
Identifying the factor pairs of the equation and checking for relatively prime conditions is crucial for solving the problem efficiently. This approach helps in quickly eliminating non-relevant pairs and focusing on the correct solution.

**Retrieved experience 2 (✗Lesson from Failure)**:
**Understanding the Role of Right Triangles in Inscribed Quadrilaterals**
The agent failed to correctly identify the right triangles formed within the quadrilateral and the specific properties of the 30-60-90 triangle, leading to an incorrect calculation of the areas. This failure underscores the need to carefully analyze the geometric properties and apply them correctly to solve problems involving inscribed quadrilaterals.

**Retrieved experience 3 (✗Lesson from Failure)**:
**Root Testing in Finite Fields**
The successful factorization of a polynomial in $Z_7[x]$ can be determined by finding its roots in the field. Testing each element of $Z_7$ (0 through 6) is necessary to identify the correct linear factors.

---

We compare the options by checking which tobacco forms are known carcinogens. Cigarettes, chewing tobacco, and snuff are all associated with cancer risk. Option D includes all three, making it the most complete list.
#### D

*Table 39.* **LLM-R's case study in Math.**

**Question**: In January the families visiting a national park see animals 26 times. In February the families that visit the national park see animals three times as many as were seen there in January. Then in March the animals are shyer and the families who visit the national park see animals half as many times as they were seen in February. How many times total did families see an animal in the first three months of the year?

**Ground Truth**: 143

**Retrieved experience 1 (✗Lesson from Failure)**:
**Identifying and Summarizing Pairs of Roots**
The key insight is to identify all pairs of positive integers whose product is 16, which helps in determining the distinct values of $k$ using Vieta's formulas. This approach ensures that all possible combinations are considered to find the average of $k$.

**Retrieved experience 2 (✗Lesson from Failure)**:
**Inverse Calculation in a Group with a Defined Operation**
The inverse of $a$ in the group $(\mathbb{Z}, *)$ is $-a - 1$. This insight is derived from the operation $a * b = a + b + 1$ and the requirement that $a * a^{-1} = 0$.

**Retrieved experience 3 (✗Lesson from Failure)**:
**Magnetic Field and Electron Path Radius**
Understanding the relationship between the magnetic field strength, electron velocity, and the radius of the circular path is crucial for predicting the path of charged particles in magnetic fields. This insight helps in accurately calculating the radius for any given conditions.

To solve this problem, we will follow the steps outlined in the experience and calculate the number of times families saw animals in each month.
1. January:
   - Families saw animals 26 times.
   - Let $J = 26$.
2. February:
   - Families saw animals three times as many as in January.
   - Therefore, $F = 3 \times J = 3 \times 26 = 78$.
3. March:
   - Families saw animals half as many times as in February.
   - Therefore, $M = \frac{F}{2} = \frac{78}{2} = 39$.
4. Total for the first three months:
   - Sum the number of times animals were seen in January, February, and March.
   - Total $= J + F + M = 26 + 78 + 39$.
Now, let's perform the addition:
   $26 + 78 = 104$
   $104 + 39 = 143$
Thus, the total number of times families saw animals in the first three months of the year is:
#### 143

*Table 40.* **LLM-R's case study in Coding.**

---

**Question**: Task: Return True if all numbers in the list `l` are below threshold `t`.
```
>>> below_threshold([1, 2, 4, 10], 100) → True
>>> below_threshold([1, 20, 4, 10], 5) → False
```
Tests:
```
assert below_threshold([1, 2, 4, 10], 100)
assert not below_threshold([1, 20, 4, 10], 5)
assert below_threshold([1, 20, 4, 10], 21)
```

---

**Ground Truth**:
```
def below_threshold(l:  list, t:  int):
  for e in l:
    if e >= t:
      return False
  return True
```

---

**Retrieved experience 1 (✗Lesson from Failure)**:
**Parallel Planes Distance Formula**
Recognizing that the planes are parallel and using the distance formula between parallel planes is crucial for solving the problem. This formula simplifies the process of finding the distance between the planes by directly applying the coefficients and constants from their equations.

**Retrieved experience 2 (✓Success Strategy)**:
**Polynomial Derivative Formula**
The insight is that to find the derivative of a polynomial, we need to multiply each coefficient by its corresponding power (index) and then shift the resulting coefficients to the left.

**Retrieved experience 3 (✗Lesson from Failure)**:
**Energy Eigenvalues and Eigenfunctions for Quantum Harmonic Oscillator**
Understanding the energy eigenvalues and the number of linearly independent eigenfunctions for a quantum harmonic oscillator is crucial for solving problems involving harmonic potentials. The energy of the third excited state is $\frac{7}{2}\hbar\omega$ and there are 3 linearly independent eigenfunctions for this energy eigenvalue.

---

####
```python
def below_threshold(l:  list, t:  int) -> bool:
  for number in l:
    if number >= t:
      return False
  return True
```

*Table 41.* **LLM-R's case study in TDC.**

**Question**: Predict the Human Intestinal Absorption property of the following molecule.
Molecule Name: Oxazepam.mol
SMILES: `O=C1Nc2ccc(Cl)cc2C(c2ccccc2)=N[C@H]1O`
Question: Will this molecule be absorbed in the human intestine?
- Positive means: High absorption (HIA+)
- Negative means: Low absorption (HIA-)

**Ground Truth**: Positive

**Retrieved experience 1 (✓Success Strategy)**:
\*\*Intestinal Absorption Prediction Insights\*\*
The molecule's moderate hydrophobicity, aromatic character, and presence of a quaternary ammonium group suggest that it is likely to be absorbed in the human intestine.
**Retrieved experience 2 (✓Success Strategy)**:
\*\*Key Factors for High Intestinal Absorption\*\*
The analysis of Propafenone's molecular structure reveals that it contains hydrophilic functional groups like carboxyl and phenolic hydroxyl groups, moderate hydrophobicity indicated by its LogP value, aromatic rings, and hydrogen bond acceptors. These factors collectively suggest that Propafenone is likely to be highly absorbed in the human intestine.
**Retrieved experience 3 (✓Success Strategy)**:
\*\*Intestinal Absorption Prediction Insights\*\*
The presence of a carboxyl group, an amine group, and a cyclohexane ring in Meperidine, along with moderate hydrophobicity and a balance of hydrogen bond donors and acceptors, suggest that the molecule is likely to be absorbed in the human intestine.

`<think>` To predict the human intestinal absorption (HIA) of Oxazepam, we need to consider the molecular structure and functional groups present in the molecule. Oxazepam is a benzodiazepine derivative, which is known to have a high potential for intestinal absorption.
1. **Pharmacophore**: The molecule contains a benzodiazepine core, which is a common structure for drugs that are well-absorbed in the intestines. This core structure is known to interact with intestinal transporters and facilitate absorption.
2. **Polar-Nonpolar Balance**: The presence of the benzodiazepine core and the chlorine atom on the cyclohexyl ring suggests a balance between hydrophilic and hydrophobic regions. This balance is often favorable for intestinal absorption.
3. **LogP**: The LogP value of the molecule is typically around 1.5 to 2.0, which is within the range for drugs that are well-absorbed in the intestines. A LogP value in this range indicates that the drug is moderately lipophilic, which is conducive to absorption.
4. **Hydrogen Bond Donors and Acceptors**: The molecule has one nitrogen atom as a hydrogen bond donor and one oxygen atom as a hydrogen bond acceptor. These functional groups can interact with intestinal transporters and facilitate absorption.
5. **Molecular Weight and Size**: The molecular weight of Oxazepam is around 184.2 g/mol, which is within the range of drug-like molecules (typically 150-300 g/mol). This size is suitable for absorption through the intestinal epithelium.
Considering all these factors, the molecular structure and functional groups of Oxazepam suggest that it will be well-absorbed in the human intestine. `</think>`
`<answer>` Positive `</answer>`

*Table 42.* **LLM-R's case study in Movie Ranking Scenario.**

**Question**: Please rank the following movies based on the likelihood that I would like to watch them next, according to my previous viewing history.

Movies Watched: ['0. Cool Hand Luke', '1. Rain Man', '2. Night of the Living Dead', '3. North by Northwest', '4. Little Big Man', '5. Butch Cassidy and the Sundance Kid', '6. Dances with Wolves', '7. High Noon', '8. Fistful of Dollars, A', '9. Blazing Saddles', '10. Hud', '11. Angel and the Badman', '12. Magnum Force', '13. Outlaw Josey Wales, The', '14. Unforgiven', '15. Jeremiah Johnson', '16. Good, The Bad and The Ugly, The', '17. High Plains Drifter', '18. Pale Rider', '19. Legends of the Fall']

Candidate Movies: ['0. Fried Green Tomatoes', '1. Hang 'em High', '2. Kazaam', '3. Kiss the Girls', '4. Little Rascals, The', '5. Space Cowboys', '6. Unstrung Heroes', '7. Golden Earrings', '8. Roadside Prophets', '9. What Lies Beneath', '10. Where the Money Is', '11. Blue Hawaii', '12. Heat', '13. Bloodsport 2', '14. Thelma & Louise', '15. Big Trouble in Little China', '16. Messenger: The Story of Joan of Arc, The', '17. Frances', '18. Bad Lieutenant', '19. Fall']

**Ground Truth**: Hang 'em High

**Retrieved experience 1 (✗Lesson from Failure)**:
**Preference for Classical Music**
Understanding and leveraging the user's preference for classical music is crucial for accurately predicting future purchases. Future recommendations should focus on classical music albums to increase the likelihood of the user's interest and satisfaction.

**Retrieved experience 2 (✓Success Strategy)**:
**Genre Preference**
The agent learned that the user frequently purchases items from specific genres like 'West Coast Jazz' and 'Rockabilly Cowboys: 250 Original Recordings', indicating a strong preference for these genres. This insight suggests that the next item should be from one of these genres to maximize the likelihood of the user's purchase.

**Retrieved experience 3 (✓Success Strategy)**:
**Genre Preference**
Based on the user's purchase history, albums with the title 'Greatest Hits' are the most frequently purchased and likely to be the next choice. This preference suggests that the user enjoys compilations of their favorite songs.

`<think>` To rank the given candidate movies based on the user's past movie preferences, let's consider the genres and themes of the movies the user has previously watched:

1. **Genre Analysis**:
- **Westerns**: 'North by Northwest', 'High Noon', 'High Plains Drifter', 'Fall'
- **Drama**: 'Cool Hand Luke', 'Rain Man', 'Little Big Man', 'Hud', 'Jeremiah Johnson', 'Unforgiven', 'Bloodsport 2', 'Fall'
- **Comedy**: 'Blazing Saddles', 'Blue Hawaii', 'Big Trouble in Little China'

...
2. **Past Movie Distribution**:
- The user has watched movies from multiple genres, indicating a balanced interest in different types of films.
3. **Candidate Movies**:
- 'Fried Green Tomatoes' (Adventure, Comedy)
- 'Hang 'em High' (Western)
- 'Kazaam' (Animation, Adventure)
- 'Kiss the Girls' (Action/Thriller)
...
4. **Ranking**:
- **Adventure**: 'Fried Green Tomatoes', 'Kazaam', 'Little Rascals, The', 'Space Cowboys'
- **Drama**: 'Little Big Man', 'Unforgiven', 'Where the Money Is', 'Fall'
- **Action/Thriller**: 'Kiss the Girls', 'Space Cowboys', 'Where the Money Is'
- **Comedy**: 'Fried Green Tomatoes', 'Little Rascals, The'
- **Western**: 'Hang 'em High', 'Fall'
...
5. **Final Ranking**:
- **1. Hang 'em High (Western)**
- **2. Fried Green Tomatoes (Adventure, Comedy)**
- **3. Kazaam (Animation, Adventure)**
- **4. Little Rascals, The (Comedy)**
- **5. Space Cowboys (Action/Thriller, Sci-Fi)**
... `</think>`
`<answer>` Hang 'em High, Fried Green Tomatoes, Kazaam, Little Rascals, The, Space Cowboys, Unforgiven, Where the Money Is, Little Big Man, Fall, Kiss the Girls `</answer>`

*Table 43.* **IRCoT's case study in QA.**

**Question**: A large, solid spherical body in the solar system is classified as a moon. Which characteristic of the body gives it this classification?
A. It rotates on its axis.
B. It lacks liquid water.
C. It orbits a nearby planet.
D. It reflects light from a star.

**Ground Truth**: C

**Retrieved experience 1 (✗Lesson from Failure)**:
**Moon's Synchronous Rotation**
Understanding the Moon's synchronous rotation is crucial for accurately calculating how many times the Moon revolves around Earth during one Earth revolution around the Sun, as it affects the apparent position of the Moon in the sky.

**Retrieved experience 2 (✗Lesson from Failure)**:
**Moon's Synchronous Rotation**
Understanding the Moon's synchronous rotation is crucial for accurately calculating how many times the Moon revolves around Earth during one Earth revolution around the Sun, as it affects the apparent position of the Moon in the sky.

**Retrieved experience 3 (✓Success Strategy)**:
**Earth's Rotation**
The Earth's rotation from west to east, completing one full rotation every 24 hours, is the primary reason why the apparent position of celestial objects like stars changes as viewed from a fixed location on Earth.

**Retrieved experience 4 (✗Lesson from Failure)**:
**Water Absence on the Moon's Surface**
The moon does not have liquid water on its surface, despite evidence of water ice in permanently shadowed craters at the lunar poles. This absence of liquid water is a key factor in the moon's environmental conditions and challenges for potential human habitation.

**Retrieved experience 5 (✗Lesson from Failure)**:
**Water Absence on the Moon's Surface**
The moon does not have liquid water on its surface, despite evidence of water ice in permanently shadowed craters at the lunar poles. This absence of liquid water is a key factor in the moon's environmental conditions and challenges for potential human habitation.

A is not defining since many bodies rotate. B is not defining since liquid water is not a classification criterion for moons. D is not defining since many bodies reflect starlight. C is defining because a moon is a natural satellite that orbits a planet.
#### C

*Table 44.* **IRCoT's case study in Math.**

**Question**: Jenny is scraping gum off the bottom of the seats in a stadium with two sections. Section A has 1 subsection with 60 seats and 3 subsections with 80 seats each. Section B has 20 more seats than 3 times as many seats as Section A has total. How many seats does Section B have?

**Ground Truth**: 920

**Retrieved experience 1 (✘Lesson from Failure)**:
\*\*Section Capacity Calculation\*\*
To accurately determine the total capacity of the baseball park, it is crucial to correctly apply the relationships between the number of seats in each section. Specifically, the blue section seats 20 fewer people than the red section, and the green section seats twice as many people as the blue section. This requires careful attention to the sequence of calculations and the relationships between the numbers.

**Retrieved experience 2 (✘Lesson from Failure)**:
\*\*Section Capacity Calculation\*\*
To accurately determine the total capacity of the baseball park, it is crucial to correctly apply the relationships between the number of seats in each section. Specifically, the blue section seats 20 fewer people than the red section, and the green section seats twice as many people as the blue section. This requires careful attention to the sequence of calculations and the relationships between the numbers.

**Retrieved experience 3 (✘Lesson from Failure)**:
\*\*Section Capacity Calculation\*\*
To accurately determine the total capacity of the baseball park, it is crucial to correctly apply the relationships between the number of seats in each section. Specifically, the blue section seats 20 fewer people than the red section, and the green section seats twice as many people as the blue section. This requires careful attention to the sequence of calculations and the relationships between the numbers.

**Retrieved experience 4 (✘Lesson from Failure)**:
\*\*Section Capacity Calculation\*\*
To accurately determine the total capacity of the baseball park, it is crucial to correctly apply the relationships between the number of seats in each section. Specifically, the blue section seats 20 fewer people than the red section, and the green section seats twice as many people as the blue section. This requires careful attention to the sequence of calculations and the relationships between the numbers.

To solve this problem, we need to follow these steps:
1. Calculate the total number of seats in Section A.
2. Use the total number of seats in Section A to find the total number of seats in Section B.
3. Provide the final answer.
### Step 1: Calculate the total number of seats in Section A
Section A has:
- 1 subsection with 60 seats.
- 3 subsections with 80 seats each.
First, calculate the total seats in the 3 subsections: $3 \times 80 = 240$
Now, add the seats from the 1 subsection: $60 + 240 = 300$
So, Section A has a total of 300 seats.
### Step 2: Calculate the total number of seats in Section B
Section B has 20 more seats than 3 times the total number of seats in Section A.
First, calculate 3 times the total number of seats in Section A: $3 \times 300 = 900$
Now, add 20 seats to this amount: $900 + 20 = 920$
So, Section B has 920 seats.
### Final Answer
#### 920

*Table 45.* **IRCoT's case study in Coding.**

**Question**: Task: The Fib4 number sequence is a sequence similar to the Fibonacci sequence defined as follows:

```
fib4(0) -> 0
fib4(1) -> 0
fib4(2) -> 2
fib4(3) -> 0
fib4(n) -> fib4(n-1) + fib4(n-2) + fib4(n-3) + fib4(n-4).
```
Do not use recursion.
```
>>> fib4(5) → 4
>>> fib4(6) → 8
>>> fib4(7) → 14
```
Tests:
```
assert fib4(5) == 4
assert fib4(8) == 28
assert fib4(10) == 104
```

**Ground Truth**:
```
def fib4(n:  int):
  results = [0, 0, 2, 0]
  if n < 4:
    return results[n]

  for _ in range(4, n + 1):
    results.append(results[-1] + results[-2] + results[-3] +
results[-4])
    results.pop(0)

  return results[-1]
```

**Retrieved experience 1 (✗Lesson from Failure)**:
**Using a Loop to Compute and Sum Fourth Powers of Odd Numbers**
To solve the problem of finding the sum of the fourth powers of the first n odd natural numbers, it is essential to use a loop to iterate through the odd numbers, compute their fourth powers, and accumulate the sum.

####
```python
def fib4(n:  int):
  """The Fib4 number sequence is a sequence similar to the Fibbonacci
sequnece that's defined as follows:
  fib4(0) -> 0
  fib4(1) -> 0
  fib4(2) -> 2
  fib4(3) -> 0
  fib4(n) -> fib4(n-1) + fib4(n-2) + fib4(n-3) + fib4(n-4).
  Please write a function to efficiently compute the n-th element of
the fib4 number sequence.  Do not use recursion.
  >>> fib4(5)
  4
  >>> fib4(6)
  8
  >>> fib4(7)
  14
  """
  results = [0, 0, 2, 0]
  if n < 4:
    return results[n]
  for _ in range(4, n + 1):
    results.append(results[-1] + results[-2] + results[-3] +
results[-4])
    results.pop(0)
  return results[-1]
```

*Table 46.* **IRCoT's case study in TDC.**

**Question**: Predict the Ames Mutagenicity Test property of the following molecule.
Molecule Name: Drug 6643
SMILES: `CCCC=CCC=O`
Question: Is this molecule mutagenic according to Ames test?
- Positive means: Mutagenic (Ames+)
- Negative means: Non-mutagenic (Ames-)

**Ground Truth**: Positive

**Retrieved experience 1 (✓Success Strategy)**:
**Ames Mutagenicity Prediction Insights**
The presence of specific functional groups like carboxyl and nitro groups can be indicative of mutagenicity, but the absence of such alerts and the typical molecular characteristics of drug-like molecules do not definitively predict Ames mutagenicity.
**Retrieved experience 2 (✓Success Strategy)**:
**Presence of Carboxylic Acid and Aromatic Rings**
The presence of a carboxylic acid group and the presence of multiple aromatic rings in the molecule suggest that it is likely to be mutagenic, as these functional groups and structural features are commonly associated with mutagenic compounds in the Ames test.
**Retrieved experience 3 (✓Success Strategy)**:
**Presence of Reactive Functional Groups**
The presence of an ester group and a hydroxyl group in the molecular structure of Drug 5561, C=Cc1ccc(O)c(OC)c1, suggests that these functional groups could potentially form reactive intermediates, which is a concern for mutagenicity. However, the overall structure does not strongly indicate mutagenicity.

**Retrieved experience 4 (✗Lesson from Failure)**:
**Functional Groups and Hydroxyl Groups as Potential Mutagenic Sites**
The carboxyl group (-COOH) and the multiple hydroxyl groups (-OH) in the given molecule are potential mutagenic sites. These functional groups can react with other molecules and potentially lead to mutagenic reactions, making the molecule a potential mutagen according to the Ames test.

**Retrieved experience 5 (✗Lesson from Failure)**:
**Hydrophobicity and Molecular Weight as Predictors of BBB Permeability**
The bromine atom in the molecule is hydrophobic, which can hinder its ability to cross the lipid bilayer of the blood-brain barrier. Additionally, the high molecular weight of the compound suggests it may be less lipophilic and thus less likely to permeate the BBB.

`<think>` The given SMILES string represents the molecule 6-hydroxybenzoic acid, which is a derivative of benzoic acid. The structure contains a hydroxyl group (-OH) and a carboxyl group (-COOH).
1. **Functional Groups**: The presence of a hydroxyl group (-OH) and a carboxyl group (-COOH) is significant. Both of these groups are known to be mutagenic. The hydroxyl group can act as a nucleophile and the carboxyl group can act as a proton donor, both of which can interfere with DNA replication and repair mechanisms.
2. **Molecular Weight and LogP**: The molecular weight of 6-hydroxybenzoic acid is 118.11 g/mol, which is within the typical range for drug-like molecules. The LogP value is 1.4, indicating moderate hydrophobicity, which is also consistent with a drug-like molecule.
3. **Structural Alerts for Toxicity**: The molecule does not contain any known structural alerts for toxicity such as nitro groups, reactive carbonyls, or other reactive functional groups that could be problematic in the Ames test.
4. **Ring Systems and Aromatic Character**: The molecule contains a benzene ring, which is a common aromatic ring system. The presence of this ring system does not inherently make the molecule mutagenic, but it is part of the overall structure.
5. **Hydrogen Bond Donors and Acceptors**: The molecule has 2 hydrogen bond donors (the carboxyl group and the nitrogen atom bonded to a carbon atom) and 1 hydrogen bond acceptor (the hydroxyl group). Given the presence of a carboxyl group and a hydroxyl group, which are known to be reactive functional groups, and considering the moderate LogP value, the molecule could potentially be mutagenic. However, the presence of a nitrogen atom bonded to a carbon atom (N-C) and a nitrogen atom bonded to a hydroxyl group (N-O) does not strongly indicate toxicity. `</think>`
`<answer>` Positive `</answer>`

*Table 47.* **IRCoT's case study in Movie Ranking Scenario.**

**Question**: Please rank the following movies based on the likelihood that I would like to watch them next, according to my previous viewing history.
Movies Watched: ['0. Star Wars: Episode I - The Phantom Menace', '1. Run Lola Run', '2. Glory', '3. Henry: Portrait of a Serial Killer', '4. Pulp Fiction', '5. Big Sleep, The', '6. East-West', '7. His Girl Friday', '8. Stranger Than Paradise', '9. Repo Man', '10. Chicken Run', '11. Rosemary's Baby', '12. High Fidelity', '13. Edward Scissorhands', '14. Bob Roberts', '15. American Beauty', '16. Not One Less', '17. Stand by Me', '18. Kronos', '19. Fantasia']
Candidate Movies: ['0. Fabulous Baker Boys, The', '1. Giant', '2. Buffy the Vampire Slayer', '3. Bride of Frankenstein', '4. Tora! Tora! Tora!', '5. No Mercy', '6. Photographer', '7. Ipcress File, The', '8. Ordinary People', '9. Paradise Road', '10. It Came from Outer Space', '11. Two Friends', '12. Goofy Movie, A', '13. When We Were Kings', '14. Escape from New York', '15. With Honors', '16. Kissing a Fool', '17. Land Before Time III: The Time of the Great Giving', '18. Slumber Party Massacre, The', '19. Sticky Fingers of Time, The']

**Ground Truth**: Being John Malkovich

**Retrieved experience 1 (✗Lesson from Failure)**:
**Ranking of Candidate Movies**
Based on the user's past preferences, the most likely next movies to watch are 'Tombstone', 'Being John Malkovich', and 'Pulp Fiction'. These movies are either in the user's watched list or are in the same genre, increasing the likelihood of the user choosing them.

**Retrieved experience 2 (✗Lesson from Failure)**:
**Frequency-Based Movie Ranking**
When ranking movies based on the likelihood of the user watching them next, we should consider the frequency of the movies in the given list. Since the user's past list does not contain any of the candidate movies, we can rank them based on their frequency in the provided list. This approach is generalizable for future tasks where the user's past list does not match the candidate list.

**Retrieved experience 3 (✓Success Strategy)**:
**Low Frequency Movies in User's Watching History**
The user's past movie list does not contain any of the candidate movies, indicating that these movies are not part of the user's preferred genre or style. Therefore, the likelihood of the user liking any of these movies next is very low.

`<think>` To rank the given candidate movies based on the user's past movie preferences, we need to analyze the user's preferences from their past movie list. Here's the step-by-step reasoning:
1. **Identify Common Genres and Themes**:
- The user has a diverse range of genres in their viewing history, including drama (e.g., 'Pulp Fiction', 'American Beauty'), thriller (e.g., 'Run Lola Run', 'Henry: Portrait of a Serial Killer'), comedy (e.g., 'His Girl Friday', 'Chicken Run'), and horror (e.g., 'Rosemary's Baby').
- This indicates that the user enjoys various genres, which will influence the ranking of candidate movies.
2. **Ranking the Candidate Movies**:
- None of the candidate movies appear in the user's past watched list. Therefore, we must use genre and thematic similarity to rank them.
- **Action/Thriller**: 'Escape from New York', 'Kissing a Fool', 'Ipcress File, The'
- **Drama**: 'Ordinary People', 'When We Were Kings', 'With Honors'
...
3. **Final Ranking**:
Given the lack of overlap with past movies, we rank the movies based on how likely they are to align with the user's broad genre preferences.
- **1. Being John Malkovich** (Drama, Comedy - Genre alignment with 'Pulp Fiction' and 'American Beauty')
- **2. Ordinary People** (Drama - Similar to the user's interest in dramas like 'Stand by Me' and 'American Beauty')
- **3. Escape from New York** (Action/Thriller - Similar to user's thrillers like 'Run Lola Run' and 'Repo Man')
- **4. Kissing a Fool** (Comedy, Drama - Similar to 'American Beauty' and 'Pulp Fiction')
- **5. Ipcress File, The** (Action/Thriller - Similar to 'Henry: Portrait of a Serial Killer')
... `</think>`
`<answer>` Being John Malkovich, Ordinary People, Escape from New York, Kissing a Fool, Ipcress File, The, Goofy Movie, A, When We Were Kings, Land Before Time III, The Time of the Great Giving, It Came from Outer Space, Slumber Party Massacre, The `</answer>`

*Table 48.* **Search-o1's case study in QA.**

**Question**: Which conclusion should the students make from the information in the table?
A. There are more plant-like organisms.
B. There are more animal-like organisms.
C. The plant-like organisms cannot move by themselves.
D. The animal-like organisms eat the plant-like organisms.

**Ground Truth**: A

```
<|begin_search_query|>
table information for organisms
<|end_search_query|>
<|begin_search_result|>
| Organism Type | Description |
|---------------|-------------|
| Plant-like    | stationary, photosynthetic |
| Animal-like   | mobile, heterotrophic |
<|end_search_result|>
```

From the table, plant-like organisms are described as "stationary, photosynthetic" and animal-like organisms as "mobile, heterotrophic."
- Option A is incorrect because the table does not compare the number of plant-like vs. animal-like organisms.
- Option B is incorrect for the same reason as A.
- Option C is correct because plant-like organisms are explicitly described as "stationary."
- Option D is incorrect because the table does not state that animal-like organisms eat plant-like organisms.

#### [C]

*Table 49.* **Search-o1's case study in Math.**

**Question**: It takes Daphne 18 minutes to finish a brain teaser and 4 minutes to finish a jigsaw puzzle. Over the day off she solved 7 brain teasers and 12 jigsaw puzzles. How much time did she spend playing these games?

**Ground Truth**: 174

To solve this problem, we need to calculate the total time Daphne spent on brain teasers and jigsaw puzzles separately, and then sum these times.
1. Calculate the time spent on brain teasers:
   - Daphne takes 18 minutes to finish one brain teaser.
   - She solved 7 brain teasers.
   - Total time for brain teasers = 18 minutes/brain teaser $\times$ 7 brain teasers.
2. Calculate the time spent on jigsaw puzzles:
   - Daphne takes 4 minutes to finish one jigsaw puzzle.
   - She solved 12 jigsaw puzzles.
   - Total time for jigsaw puzzles = 4 minutes/jigsaw puzzle $\times$ 12 jigsaw puzzles.
3. Sum the total time spent:
   - Total time = Total time for brain teasers + Total time for jigsaw puzzles.
Let's perform the calculations:

```
<|begin_search_query|>
```
total time for brain teasers = 18 * 7
total time for jigsaw puzzles = 4 * 12
total time = total time for brain teasers + total time for jigsaw puzzles
```
<|end_search_query|>
<|begin_search_result|>
```
Final Information
total time for brain teasers = 18 * 7
total time for jigsaw puzzles = 4 * 12
total time = total time for brain teasers + total time for jigsaw puzzles
Let's perform the calculations:
- Total time for brain teasers = $18 \times 7 = 126$ minutes
- Total time for jigsaw puzzles = $4 \times 12 = 48$ minutes
- Total time = $126 + 48 = 174$ minutes

Daphne spent a total of 174 minutes playing brain teasers and jigsaw puzzles.
```
<|end_search_result|>
```
#### 174

*Table 50.* **Search-o1's case study in Coding.**

**Question**: Task: Write a function that takes a string as input and returns the sum of the upper characters only' ASCII codes.
Examples:
```
digitSum("") => 0
digitSum("abAB") => 131
digitSum("abcCd") => 67
digitSum("helloE") => 69
digitSum("woArBld") => 131
digitSum("aAaaaXa") => 153
```
Function signature:
```
def digitSum(s):
```
Tests:
```
assert True, "This prints if this assert fails 1 (good for debugging!)"
assert digitSum("") == 0, "Error"
assert digitSum("abAB") == 131, "Error"
```

**Ground Truth**:
```
def digitSum(s):
  if s == "":  return 0
  return sum(ord(char) if char.isupper() else 0 for char in s)
```

Assistant thinking steps:
- The task is to sum the ASCII values of uppercase letters in a string.
- We need to iterate through each character, check if it is an uppercase letter, and if so, add its ASCII value to a running total.

Assistant:
```
<|begin_search_query|>python sum ascii values of uppercase letters in a
string<|end_search_query|>
<|begin_search_result|>**Inputs:**
```
- **Previous Reasoning Steps:**
Assistant thinking steps:
- The task is to sum the ASCII values of uppercase letters in a string.
- We need to iterate through each character, check if it is an uppercase letter, and if so, add its ASCII value to a running total.
```
<|end_search_result|>
```

####
```python
def digitSum(s):
  total = 0
  for char in s:
    if char.isupper():
      total += ord(char)
  return total
```

*Table 51.* **Search-o1's case study in TDC.**

---

**Question**: Predict the Blood-Brain Barrier Penetration property of the following molecule.
Molecule Name: eticyclidine
SMILES: `CCNC1(c2ccccc2)CCCCC1`
Question: Can this molecule cross the blood-brain barrier?
- Positive means: BBB permeable (BBB+)
- Negative means: BBB impermeable (BBB-)

---

**Ground Truth**: Positive

---

`<|begin_search_query|>`Blood-Brain Barrier Penetration eticyclidine BBB permeability functional groups aromaticity hydrophobicity LogP`<|end_search_query|>`
`<|begin_search_result|>`
**Final Information**
**experience 1 (✓Success Strategy):**
The molecule contains functional groups like carbonyl and carboxyl, moderate hydrophobicity, and a balanced number of hydrogen bond donors and acceptors, suggesting it is likely to be BBB permeable.
- **Functional Groups:** Carbonyl and carboxyl
- **Hydrophobicity:** Moderate
- **LogP:** Balanced (implied from moderate hydrophobicity)
Given the previous reasoning steps focused on BBB permeability and the provided information, we can conclude that eticyclidine likely has moderate BBB permeability due to its functional groups and hydrophobicity.
`<|end_search_result|>`
`<think>` From the search results, we understand that eticyclidine contains functional groups like carbonyl and carboxyl, which are hydrophobic. The molecule also has moderate hydrophobicity and a balanced number of hydrogen bond donors and acceptors. These characteristics suggest that the molecule is likely to be BBB permeable. `</think>`
`<answer>` Positive `</answer>`

---

*Table 52.* **Search-o1's case study in Movie Ranking Scenario.**

**Question**: Please rank the following movies based on the likelihood that I would like to watch them next, according to my previous viewing history.

Movies Watched: ['0. Meatballs III', '1. Back to School', '2. From Here to Eternity', '3. L.A. Confidential', '4. White Men Can't Jump', '5. Fatal Instinct', '6. Dial M for Murder', '7. 20 Dates', '8. Arachnophobia', '9. 10 Things I Hate About You', '10. American Beauty', '11. Abyss, The', '12. American Pie', '13. Arlington Road', '14. Austin Powers: The Spy Who Shagged Me', '15. Battlefield Earth', '16. Being John Malkovich', '17. Beach, The', '18. Bug's Life, A', '19. Blair Witch Project, The']

Candidate Movies: ['0. 52 Pick-Up', '1. Nico Icon', '2. Saturn 3', '3. Lord of the Flies', '4. Cat from Outer Space, The', '5. Bank Dick, The', '6. Cops and Robbersons', '7. Railroaded!', '8. Rawhead Rex', '9. Dead Ringers', '10. My Tutor', '11. Chuck & Buck', '12. Shower', '13. Jackie Brown', '14. Blues Brothers 2000', '15. Love and Other Catastrophes', '16. Police Academy 3: Back in Training', '17. Wayne's World 2', '18. Prince of the City', '19. Everyone Says I Love You']

**Ground Truth**: Cat from Outer Space, The

<|begin_search_query|>
Please rank the following movies based on my watching history:...
<|end_search_query|>
<|begin_search_result|>
*Final Information**
Alphabetical Ranking Insight
Since the user's past movie preferences do not provide a clear indication of future choices, the agent ranks the candidate movies alphabetically....<|end_search_result|>
<think> To rank the given candidate movies based on the user's past movie preferences, we need to analyze the user's preferences from their past movie list. Here's the step-by-step reasoning:
1. **Identify Common Genres and Themes**:
- The user has a diverse range of genres in their viewing history, including drama (e.g., 'Pulp Fiction', 'American Beauty'), thriller (e.g., 'Run Lola Run', 'Henry: Portrait of a Serial Killer'), comedy (e.g., 'His Girl Friday', 'Chicken Run'), and horror (e.g., 'Rosemary's Baby').
- This indicates that the user enjoys various genres, which will influence the ranking of candidate movies.
2. **Ranking the Candidate Movies**:
- None of the candidate movies appear in the user's past watched list. Therefore, we must use genre and thematic similarity to rank them.
- **Action/Thriller**: 'Escape from New York', 'Kissing a Fool', 'Ipcress File, The'
- **Drama**: 'Ordinary People', 'When We Were Kings', 'With Honors'

...
3. **Final Ranking**:
Given the lack of overlap with past movies, we rank the movies based on how likely they are to align with the user's broad genre preferences.
- **1. Cat from Outer Space, The (Adventure, Family)** - Similar to the user's interest in family films such as 'Bug's Life, A'.
- **2. 52 Pick-Up (Action/Thriller)** - Matches the user's interest in thrillers like 'L.A. Confidential'.
- **3. Bank Dick, The (Comedy)** - Aligns with comedy themes like 'Austin Powers: The Spy Who Shagged Me'.
- **4. Cops and Robbersons (Comedy, Crime)** - Similar to 'Pulp Fiction' in its mix of humor and crime.
- **5. Railroaded! (Drama)** - Matches with the user's interest in crime and drama films like 'American Beauty'.
... </think>
<answer> Cat from Outer Space, The, 52 Pick-Up, Bank Dick, The, Cops and Robbersons, Railroaded!, Rawhead Rex, Dead Ringers, My Tutor, Chuck & Buck, Shower, Jackie Brown, Blues Brothers 2000, Love and Other Catastrophes, Police Academy 3: Back in Training, Wayne's World 2, Prince of the City, Everyone Says I Love You </answer>

*Table 53.* **R1's case study in QA.**

**System Prompt**: You are a helpful assistant skilled in mathematics, logical reasoning, and programming. Solve problems step by step, showing your work clearly. You MUST end your response with '#### ' followed by your final answer.

**User Prompt**: Solve this multiple-choice question:

A bowling ball with a mass of 8.0 kg rolls down a bowling lane at 2.0 m/s. What is the momentum of the bowling ball?
A. 4.0 kg x m/s
B. 6.0 kg x m/s
C. 10.0 kg x m/s
D. 16.0 kg x m/s

Think step by step, then write your final answer as: #### [letter]

**Ground Truth**: D

To find the momentum of the bowling ball, we use the formula for momentum, which is given by:

$$\text{Momentum} = \text{mass} \times \text{velocity}$$

Given:
- Mass of the bowling ball = 8.0 kg
- Velocity of the bowling ball = 2.0 m/s

Substituting the given values into the formula:

$$\text{Momentum} = 8.0\,\text{kg} \times 2.0\,\text{m/s} = 16.0\,\text{kg} \cdot \text{m/s}$$

Now, let's analyze the options:
A. 4.0 kg x m/s
B. 6.0 kg x m/s
C. 10.0 kg x m/s
D. 16.0 kg x m/s

The correct answer is 16.0 kg x m/s, which corresponds to option D.

#### D

*Table 54.* **R1's case study in Math.**

**System Prompt**: You are a helpful assistant skilled in mathematics, logical reasoning, and programming. Solve problems step by step, showing your work clearly. You MUST end your response with '#### ' followed by your final answer.

**User Prompt**: Solve this math problem:

Olga has an aquarium with fish in 5 different colors. There are 12 yellow ones, half as many blue ones, and twice as many green ones as yellow ones. How many fish does Olga have in her aquarium?

Think step by step, showing your work clearly. Write your final answer as: #### [answer]

**Ground Truth**: 42

To determine the total number of fish in Olga's aquarium, we need to calculate the number of fish of each color and then sum them up.

1. **Yellow Fish:** Olga has 12 yellow fish.

2. **Blue Fish:** The number of blue fish is half the number of yellow fish. Therefore, the number of blue fish is:
$\frac{12}{2} = 6$

3. **Green Fish:** The number of green fish is twice the number of yellow fish. Therefore, the number of green fish is:
$12 \times 2 = 24$

4. **Calculating the Total Number of Fish:** Now, we sum the number of fish of each color:
$12 \text{ (yellow)} + 6 \text{ (blue)} + 24 \text{ (green)} = 42$

Thus, the total number of fish in Olga's aquarium is 42.

#### 42

*Table 55.* **R1's case study in Coding.**

**System Prompt**: You are an expert Python programmer. Solve coding problems step by step. You MUST end your response with '#### ' followed by your complete code in a "'python code block.

**User Prompt**: Task: Given an array arr of integers, find the minimum number of elements that need to be changed to make the array palindromic.

For example:
smallest_change([1,2,3,5,4,7,9,6]) == 4
smallest_change([1, 2, 3, 4, 3, 2, 2]) == 1
smallest_change([1, 2, 3, 2, 1]) == 0

Think through your approach, then write the Python function. REMEMBER: You MUST end with ####
followed by your code in a "'python block.

**Ground Truth**:
```
def smallest_change(arr):
  ans = 0
  for i in range(len(arr) // 2):
    if arr[i] != arr[len(arr) - i - 1]:
      ans += 1
  return ans
```

To solve this problem, we can leverage the fact that a palindromic array reads the same forwards and backwards.

Here's a step-by-step plan:
1. Initialize a counter to keep track of the number of changes needed.
2. Use two pointers, one starting from the beginning and the other from the end of the array.
3. Compare the elements at these pointers. If they are not equal, increment the change counter.
4. Move the pointers towards the center.
5. Return the total number of changes.

```
####
def smallest_change(arr):
  changes = 0
  left, right = 0, len(arr) - 1
  while left < right:
    if arr[left] != arr[right]:
      changes += 1
    left += 1
    right -= 1
  return changes
```

*Table 56.* **R1's case study in TDC.**

**System Prompt**: You are a helpful assistant skilled in chemistry and molecular property prediction. Analyze molecules step by step, showing your reasoning clearly. You MUST end your response with '#### ' followed by your final answer (Positive or Negative).

**User Prompt**: Predict the Ames Mutagenicity Test property of the following molecule.

Molecule Name: Drug 1955
SMILES: CN(C#N)N=O

Question: Is this molecule mutagenic according to Ames test?
- Positive means: Mutagenic (Ames+)
- Negative means: Non-mutagenic (Ames-)

Think step by step, then write your final answer as: #### [Positive/Negative]

**Ground Truth**: Positive

<think>
The given molecule is Drug 1955 with the SMILES representation CN(C#N)N=O. This structure contains a nitroso group (N=O) which is a known mutagenic functional group. Nitroso compounds are often recognized as potential mutagens due to their ability to form covalent bonds with DNA, leading to mutations. Additionally, the presence of a cyano group (C#N) suggests that the molecule could have reactive sites that might interact with DNA. These structural features make the molecule likely to be mutagenic.
</think>

#### Positive

---

*Table 57.* **R1's case study in Movie Recommendation.**

**System Prompt**: You are a helpful assistant skilled in movie recommendation. Analyze user preferences step by step, showing your reasoning clearly. You MUST end your response with '#### ' followed by your recommended movie.

**User Prompt**: Based on the user's viewing history, recommend the next movie.

Watched: ['Taxi Driver', 'The Good, The Bad and The Ugly', 'Dances with Wolves']
Candidates: ['Deer Hunter', 'Rocky', 'Rambo', 'Platoon']

Think step by step about the user's preferences, then write your final answer as: #### [movie name]

**Ground Truth**: Deer Hunter, The

<think>
Let's analyze the user's watching history to identify patterns:

1. **Taxi Driver** (1976) - A psychological drama with intense themes.
2. **The Good, The Bad and The Ugly** (1966) - Classic western epic.
3. **Dances with Wolves** (1990) - Western drama with deep character study.

The user shows preference for character-driven dramas and western films.
"Deer Hunter" combines war drama with psychological depth.
</think>

#### Deer Hunter, The

*Table 58.* **SearchR1's case study in QA (with search).**

**System Prompt**: You are a reasoning assistant with the ability to perform web searches to help you answer the user's question accurately. You can search for information when needed, and the system will return relevant results.

**User Prompt**: Please answer the following question. Think step by step. Provide your final answer in the format \boxed{YOUR_ANSWER}.

**Question**: The heart muscle is stimulated to contract by electrical impulses which are generated where?

**Ground Truth**: sinoatrial (SA) node

`<think>` I need to recall where the heart's electrical impulses originate. The heart has a specialized conduction system that controls the heartbeat rhythm. Let me search for more specific information about this. `</think>`
`<search>` origin of heart electrical impulses cardiac pacemaker `</search>`

`<information>`
Doc 1(Title: "Cardiac pacemaker") The contraction of cardiac muscle is initiated by electrical impulses known as action potentials. The cells that create these rhythmic impulses are called pacemaker cells, and they are concentrated in specialized regions of the heart. In most humans, the sinoatrial (SA) node located in the right atrium serves as the natural pacemaker, setting the pace for the entire heart.
Doc 2(Title: "Electrical conduction system") The electrical conduction system of the heart transmits signals generated by the sinoatrial node through the atria to the atrioventricular (AV) node, then to the bundle of His, and finally to the Purkinje fibers, causing coordinated contraction of the heart muscle.
`</information>`

`<think>` Based on the search results, I can confirm that the heart's electrical impulses are generated in the sinoatrial (SA) node, which is located in the right atrium. This node contains specialized pacemaker cells that spontaneously generate action potentials, setting the rhythm for the entire heart. The SA node is often called the heart's natural pacemaker. `</think>`

`<answer>` sinoatrial (SA) node `</answer>`

*Table 59.* **SearchR1's case study in Math (with search).**

**System Prompt**: You are a reasoning assistant with the ability to perform web searches to help you answer the user's question accurately. You can search for information when needed, and the system will return relevant results.

**User Prompt**: Please answer the following math question. Think step by step. Provide your final answer in the format \boxed{YOUR_ANSWER}.

**Question**: A store sells notebooks for $3 each and pens for $1.50 each. If Sarah buys 4 notebooks and some pens, spending exactly $18, how many pens did she buy?

**Ground Truth**: 4

`<think>` This is an algebraic word problem. I need to first calculate how much Sarah spent on notebooks, then determine how much money remains for pens. Let me search for a systematic approach to solve this type of problem. `</think>`
`<search>` algebraic word problem shopping calculation strategy `</search>`

`<information>`
Doc 1(Title: "Algebraic problem solving") To solve shopping problems involving multiple items: Step 1: Calculate the total cost of items with known quantities. Step 2: Subtract from the total budget to find remaining money. Step 3: Divide the remaining amount by the unit price of the unknown item to find the quantity.
`</information>`

`<think>` Following the strategy from the search results:
Step 1: Calculate cost of 4 notebooks = 4 x $3 = $12
Step 2: Remaining money = $18 - $12 = $6
Step 3: Number of pens = $6 / $1.50 = 4 pens

Let me verify: 4 notebooks ($12) + 4 pens ($6) = $18. This matches the total spent. `</think>`

`<answer>` 4 `</answer>`

*Table 60.* **SearchR1's case study in Coding (with search).**

**System Prompt**: You are a reasoning assistant with the ability to perform web searches to help you answer the user's question accurately. You can search for information when needed, and the system will return relevant results.

**User Prompt**: You will be given a question (problem specification) and will generate a correct Python program that matches the specification and passes all tests. Think step by step. Enclose your final code within Python code delimiters.

**Question**: Write a function to check if a given string is a valid palindrome, considering only alphanumeric characters and ignoring cases.

Function signature: `def is_palindrome(s:  str) -> bool:`

**Ground Truth**:
```
def is_palindrome(s):
  s = ''.join(c.lower() for c in s if c.isalnum())
  return s == s[::-1]
```

`<think>` I need to implement a palindrome checker that handles special cases: ignoring non-alphanumeric characters and being case-insensitive. Let me search for best practices on string processing for palindrome checking. `</think>`
`<search>` python palindrome alphanumeric case insensitive string processing `</search>`

`<information>`
Doc 1(Title: "String processing for palindromes") To check palindromes considering only alphanumeric characters: 1) Use isalnum() to filter characters, 2) Convert all characters to lowercase using lower(), 3) Compare the processed string with its reverse using slicing [::-1]. This approach handles edge cases like spaces, punctuation, and mixed case.
`</information>`

`<think>` Based on the search results, I'll implement the solution in three steps:
1. Filter the string to keep only alphanumeric characters using isalnum()
2. Convert to lowercase for case-insensitive comparison
3. Compare the filtered string with its reverse using Python's slice notation

This handles inputs like "A man, a plan, a canal: Panama" correctly. `</think>`

`<answer>`
```
def is_palindrome(s):
  filtered = ''.join(c.lower() for c in s if c.isalnum())
  return filtered == filtered[::-1]
```
`</answer>`

*Table 61.* **SearchR1's case study in TDC (with search).**

---

**System Prompt**: You are a reasoning assistant with the ability to perform web searches to help you answer the user's question accurately. You can search for information when needed, and the system will return relevant results.

---

**User Prompt**: Predict the molecular property based on the SMILES representation. Think step by step. Provide your final answer in the format \boxed{YOUR_ANSWER}.

---

**Question**: Predict the Blood-Brain Barrier Penetration property of the molecule.

Molecule Name: Caffeine
SMILES: `Cn1cnc2c1c(=O)n(c(=O)n2C)C`

Question: Can this molecule cross the blood-brain barrier?
- Positive means: BBB permeable (BBB+)
- Negative means: BBB impermeable (BBB-)

---

**Ground Truth**: Positive

---

`<think>` To predict BBB permeability, I need to analyze the molecular properties of caffeine. Key factors include molecular weight, lipophilicity, and hydrogen bonding capacity. Let me search for information about caffeine's pharmacokinetic properties. `</think>`
`<search>` caffeine blood brain barrier permeability molecular weight lipophilicity `</search>`

`<information>`
Doc 1(Title: "BBB permeability factors") Molecules that readily cross the blood-brain barrier typically have: molecular weight below 500 Da, moderate lipophilicity (logP 1-3), limited hydrogen bond donors (<5), and low polar surface area. Caffeine has MW of 194 Da and is a well-known CNS stimulant.
Doc 2(Title: "Caffeine pharmacology") Caffeine is rapidly absorbed and readily crosses the blood-brain barrier due to its small molecular size (194.19 Da), moderate lipophilicity, and favorable physicochemical properties. It reaches peak CNS concentrations within 30-45 minutes of oral administration.
`</information>`

`<think>` The search confirms that caffeine readily crosses the BBB. Key supporting evidence:
1. Low molecular weight (194 Da) - well below the 500 Da threshold
2. Moderate lipophilicity - favorable for membrane permeation
3. Known CNS stimulant - empirically confirmed BBB penetration
4. Rapid brain uptake - reaches peak CNS levels within 30-45 minutes

Based on these molecular and pharmacokinetic properties, caffeine is BBB permeable. `</think>`

`<answer>` Positive `</answer>`

---

*Table 62.* **SearchR1's case study in Movie Recommendation (with search).**

**System Prompt**: You are a reasoning assistant with the ability to perform web searches to help you answer the user's question accurately. You can search for information when needed, and the system will return relevant results.

**User Prompt**: Based on the user's viewing history, recommend the next movie. Think step by step. Provide your final answer in the format \boxed{YOUR_ANSWER}.

**Question**: Please rank the following movies based on the likelihood that I would like to watch them next, according to my previous viewing history.

Movies Watched: ['0. The Shawshank Redemption', '1. The Godfather', '2. Schindler's List', '3. 12 Angry Men', '4. The Dark Knight', '5. Goodfellas', '6. Se7en', '7. City of God', '8. The Usual Suspects', '9. Leon: The Professional']
Candidate Movies: ['0. Forrest Gump', '1. The Matrix', '2. Pulp Fiction', '3. Fight Club', '4. Inception', '5. The Departed', '6. Memento', '7. The Prestige', '8. American History X', '9. Reservoir Dogs']

**Ground Truth**: Pulp Fiction

`<think>` I need to analyze the user's viewing history to identify patterns and preferences. The watched movies span multiple decades and include various acclaimed films. Let me search for information about these movies and their common characteristics. `</think>`
`<search>` movies similar to Shawshank Redemption Godfather Goodfellas crime drama classic film recommendations `</search>`

`<information>`
Doc 1(Title: "Classic Crime Drama Analysis") Users who enjoyed The Godfather and Goodfellas frequently appreciate Pulp Fiction and Reservoir Dogs. These films share themes of organized crime, moral ambiguity, and feature innovative storytelling with deep character development. Quentin Tarantino's works are particularly popular among fans of Scorsese and Coppola.
Doc 2(Title: "Film Recommendation Patterns") Viewers of The Shawshank Redemption and 12 Angry Men tend to enjoy character-driven narratives with themes of justice and redemption. Pulp Fiction (1994) revolutionized crime drama with its non-linear narrative structure, sharing thematic elements with The Godfather in exploring morality within criminal contexts.
Doc 3(Title: "IMDb Top 250 Correlations") Statistical analysis shows strong viewing correlations: users who watched The Usual Suspects (93% correlation with Pulp Fiction), Se7en fans frequently enjoy Fight Club and Memento. The Dark Knight viewers often appreciate Inception and The Prestige by Christopher Nolan.
`</information>`

`<think>` Based on the search results and user's viewing history, let me analyze the patterns:

1. **Genre Preferences**: The user strongly favors crime dramas (Godfather, Goodfellas, The Usual Suspects) and psychological thrillers (Se7en, Leon). This suggests high affinity for Pulp Fiction and Reservoir Dogs.

2. **Director Patterns**: User has watched Scorsese (Goodfellas), suggesting appreciation for stylized crime narratives. Tarantino's films (Pulp Fiction, Reservoir Dogs) share similar aesthetic sensibilities.

3. **Thematic Elements**: Redemption (Shawshank), moral complexity (12 Angry Men), and character depth appear consistently. Pulp Fiction excels in all these dimensions.

4. **Critical Acclaim**: All watched films are IMDb Top 50. Among candidates, Pulp Fiction ranks highest in critical consensus and shares the most genre overlap.

5. **Ranking Decision**: Pulp Fiction best matches the user's demonstrated preferences for crime drama with complex narratives. The Departed and Reservoir Dogs are strong secondary choices given the Scorsese/crime drama pattern.
`</think>`

`<answer>` Pulp Fiction, Reservoir Dogs, The Departed, Fight Club, Memento, American History X, Inception, The Prestige, The Matrix, Forrest Gump `</answer>`

*Table 63.* **ExpWeaver**'s case study in Question Answering.

---

**Question**: Which process is most directly responsible for the transformation of kinetic energy to thermal energy?
A. a ball rolling down a ramp
B. a car braking to a stop
C. a rock falling from a cliff
D. a spring bouncing a weight

---

**Ground Truth**: B

---

```
<think>
```
To identify which process most directly converts kinetic energy to thermal energy, I need to analyze each option:
Option A (ball rolling down ramp): The ball primarily converts gravitational potential energy to kinetic energy as it descends. Friction exists but is not the dominant energy transformation.
Option B (car braking to stop): When a car brakes, the brake pads press against the rotors, creating friction. This friction directly converts the car's kinetic energy (motion) into thermal energy (heat). This is why brake discs become hot after heavy braking.
Option C (rock falling from cliff): Similar to Option A, the rock converts gravitational potential energy to kinetic energy during the fall. Air resistance generates minimal thermal energy.
Option D (spring bouncing weight): The spring stores kinetic energy as elastic potential energy and releases it back. The primary transformation is between kinetic and elastic potential energy, not thermal.
The braking process in Option B is specifically designed to dissipate kinetic energy as heat through friction, making it the most direct kinetic-to-thermal conversion.
```
</think>
<answer>B</answer>
```

---

*Table 64.* **ExpWeaver**'s case study in Math Reasoning.

---

**Question**: A farmer has 120 meters of fencing to enclose a rectangular field. If the length of the field is twice the width, what is the area of the field in square meters?

---

**Ground Truth**: 800

---

```
<think>
```
Let me solve this step by step.
1. **Define variables:**
   Let $w$ = width of the field
   Since length is twice the width: $l = 2w$
2. **Set up the perimeter equation:**
   Total fencing = Perimeter = 120 meters
   Perimeter of rectangle: $P = 2l + 2w$
   Substituting $l = 2w$: $2(2w) + 2w = 120$
   Simplifying: $4w + 2w = 120$
   Therefore: $6w = 120$
3. **Solve for width:**
   $w = 120 \div 6 = 20$ meters
4. **Calculate length:**
   $l = 2w = 2 \times 20 = 40$ meters
5. **Calculate area:**
   Area $= l \times w = 40 \times 20 = 800$ square meters
Verification: $2(40) + 2(20) = 80 + 40 = 120$ meters ✓
```
</think>
<answer>800</answer>
```

*Table 65.* **ExpWeaver**'s case study in Coding.

**Question**: Write a function that takes a list of integers and returns a new list containing only the elements that appear exactly once (unique elements), preserving their original order.

```
>>> unique_elements([1, 2, 2, 3, 4, 4, 5]) → [1, 3, 5]
>>> unique_elements([1, 1, 1]) → []
>>> unique_elements([7, 8, 9]) → [7, 8, 9]
```

**Ground Truth**:
```
def unique_elements(lst):
  from collections import Counter
  counts = Counter(lst)
  return [x for x in lst if counts[x] == 1]
```

<think>
To solve this problem, I need to:
1. Count the frequency of each element
2. Filter elements that appear exactly once
3. Preserve the original order
Key insight: Using `set()` would be incorrect—it removes duplicates but would include elements that appeared multiple times. The requirement is elements appearing *exactly once*.
Approach:
- Use `collections.Counter` for O(n) frequency counting
- Iterate through the original list (not a set) to preserve order
- Include only elements where count equals 1
This handles edge cases: empty lists return empty, all-duplicates return empty, and all-unique returns the original list.
</think>
<answer>
```
def unique_elements(lst):
  from collections import Counter
  freq = Counter(lst)
  return [x for x in lst if freq[x] == 1]
```
</answer>

*Table 66.* **ExpWeaver**'s case study in TDC.

---

**Question**: Predict the Blood-Brain Barrier Penetration property of the following molecule.

Molecule Name: Donepezil
SMILES: `COc1cc2CC(CC2cc1OC)C(=O)N1CCc2ccccc2C1`

Question: Can this molecule cross the blood-brain barrier?
- Positive means: BBB permeable (BBB+)
- Negative means: BBB impermeable (BBB-)

---

**Ground Truth**: Positive

---

`<think>`
Let me analyze the molecular structure of Donepezil for BBB permeability prediction.
**Structural Analysis from SMILES:**
- Two methoxy groups (-OCH$_3$) attached to a benzene ring
- An indanone-like core structure
- A piperidine ring (6-membered nitrogen heterocycle)
- A carbonyl group (C=O) forming an amide linkage
- A benzene ring fused to the piperidine
**Physicochemical Properties:**
- Molecular weight: $\approx$ 379 Da (well below 450 Da threshold for BBB penetration)
- Contains aromatic rings contributing to moderate lipophilicity
- Methoxy groups increase logP without adding hydrogen bond donors
- Limited polar surface area due to minimal H-bond donors
**Pharmacological Context:**
- Donepezil is a known acetylcholinesterase inhibitor
- It is an FDA-approved drug for Alzheimer's disease treatment
- As a CNS-active drug, it must cross the BBB to exert its therapeutic effect
All structural features—moderate molecular weight, favorable lipophilicity from aromatic rings and methoxy groups, and limited hydrogen bond donors—strongly support BBB permeability.
`</think>`
`<answer>`Positive`</answer>`

---

*Table 67.* **ExpWeaver**'s case study in Movie Recommendation.

---

**Question**: Based on the user's viewing history, recommend the next movie.

Watched: ['The Shawshank Redemption', 'Schindler's List', 'Forrest Gump', 'The Green Mile', 'Good Will Hunting']
Candidates: ['The Pursuit of Happyness', 'Transformers', 'Awakenings', 'Fast & Furious', 'Rain Man']

---

**Ground Truth**: The Pursuit of Happyness

---

`<think>`
Let me analyze the user's watching history to identify patterns:
1. **The Shawshank Redemption** (1994) - Drama about hope and redemption in prison.
2. **Schindler's List** (1993) - Historical drama with emotional depth.
3. **Forrest Gump** (1994) - Inspirational drama about perseverance.
4. **The Green Mile** (1999) - Emotional drama with redemption themes.
5. **Good Will Hunting** (1997) - Character-driven drama about self-discovery.
**Pattern identified:** The user strongly prefers character-driven dramas with themes of human resilience, redemption, and emotional depth. No action or animated films in history.
**Candidate evaluation:**
- **The Pursuit of Happyness**: Biographical drama about perseverance and overcoming adversity—perfect thematic match
- **Awakenings**: Medical drama with emotional depth—strong match
- **Rain Man**: Character-driven drama—good match
- **Transformers**: Action blockbuster—genre mismatch
- **Fast & Furious**: Action blockbuster—genre mismatch
"The Pursuit of Happyness" best aligns with the user's preference for inspirational biographical dramas about overcoming adversity.
`</think>`
`<answer>`The Pursuit of Happyness`</answer>`

