# OpenReview forum: "ExpWeaver: LLM Agents Learn from Experience via Latent RAG"
_ICML.cc/2026/Conference — ICML 2026 regular_

### Official Review · Reviewer_gML7 · 2026-03-09

**Soundness:** 3
**Presentation:** 3
**Significance:** 3
**Originality:** 3
**Overall Recommendation:** 5
**Confidence:** 3

**Summary:**

This paper proposes an Agentic RAG system called Expweaver. Conventional retrieval approach of previous solutions precomputes a textual corpus by using an another transformer encoder to encode and index text chunks. During inference, they explicitly utilize that transformer encoder to encode retrieval queries or questions and perform similarity matching. And finally, the retrieved contexts will be inserted back to the input sequence of the LLM to perform reasoning. Nevertheless, Expweaver adopts an alternative approach. It gathers past experiences, including the questions, answers, reasoning trajectories, rewards from previous attempts and encode them into a latent embedding based on the same LLM, and store them into an “experience bank”. Rather than retrieving raw text, Expweaver retrieves the top-k embeddings at the token level during inference. Using the hidden representation of its last layer, the model performs cross-attention and gated integration to seamlessly weave past knowledge into its latent space right prior to generating each token. The authors conduct comprehensive experiments on various tasks, namely general QA, math, coding, and recommendation systems.

**Compliance With Llm Reviewing Policy:**

Affirmed.

**Final Justification:**

In general, this paper demonstrates a complete and solid work. This paper believes most of its limitations lie in the writing part, which are indeed minor and can be resolved by the camera-ready version. Therefore, this reviewer would like to give an overall recommendation of acceptance.

**Key Questions For Authors:**

1.	This reviewer is interested in understanding how the degree of off-policy of previous experiences affects performance.

**Limitations:**

Yes.

**Strengths And Weaknesses:**

Strengths:

1.	This paper is generally clearly written and well-motivated.

2.	Explicit textual retrieval requires repetitive sequence encoding. Using the same policy model (LLM) for encoding can effectively remove such computational burden.

3.	The proposed method shows to be novel, which directly injects the knowledge into the hidden representation of tokens via cross attention and gated integration.

4.	The proposed method is broadly evaluated in different tasks, including QA, math, coding and recommendation systems.

Weaknesses:

1.	Prior to the section 2, this reviewer suggests the authors to explicitly state and define “experience” as a summarization of question, answer, reward and outcome. This reviewer believes this will help the readers to better comprehend the target of this paper.

2.	Regarding Table 1, retrieval in latent space and adaptive retrieval are two distinct concepts, but the manuscript seems to conflate them. In particular, this reviewer is not convinced that Search-R1 should be classified as non-adaptive retrieval. To the best of this reviewer’s knowledge, search agents typically adjust their planning and retrieval behavior dynamically based on intermediate feedback, rather than following a fixed rule. The paper should therefore define adaptive retrieval more precisely and clarify why Search-R1 and/or similar agents does not meet this definition.

3.	As training progresses, past off-policy experiences may become increasingly distorted relative to the current model. This issue is likely to become more pronounced as learning continues.

4.	This reviewer suggests the author so unify the notations. In line 12, algorithm 1 (line 183 left), the paper uses $s_i$ to denote the experience. However, in equation 5 (line 140 right), the notation is $e$.

---

> ### Author Rebuttal · Authors · 2026-03-31
>
> Thanks for the constructive comments. We address each point in detail below and will include these results in the revised paper.
>
> **[W1: Experience Definition and Clarity]** Agreed. We will add a formal definition of experience before Section 2.
>
> ---
>
> **[W2: Adaptive Retrieval Definition]** The reviewer is correct that **Search-R1** is adaptive in **query generation**, as it dynamically formulates sub-queries based on intermediate reasoning states. We agree that our use of **“Adaptive Retrieval”** in Table 1 was not sufficiently precise and will revise the caption accordingly. In our paper, **Adaptive Retrieval** means that the **retrieval relevance criterion itself** adapts to the model’s current generation state and is jointly optimized with the policy toward the task objective, rather than relying on a fixed external retriever with a decoupled similarity function. Under this definition, Search-R1 does not qualify: although it performs adaptive query generation, its retrieval still depends on a fixed external retriever (e.g., E5) with a non-learnable similarity function independent of the LLM’s internal state and task feedback. By contrast, **ExpWeaver** uses the model’s current hidden state as the retrieval query at each decoding step, with retrieval relevance jointly optimized through RL. We will clarify this distinction in the caption of Table 1 to avoid conflating **query-level adaptivity** with **retrieval-level adaptivity**.
>
> ---
>
> **[W3: Off-Policy Drift and Experience Reliability]** We clarify that ExpWeaver addresses
> this issue through a principled experience management design: **each experience is
> encoded with the current policy at the time of storage (Section 3.1)**, rather than
> a fixed frozen encoder. This creates an implicit recency bias—as the policy evolves,
> newly added experiences naturally reside in a more compatible embedding region relative
> to current query hidden states, causing the retrieval mechanism to intrinsically favor
> more recent, policy-aligned experiences. Empirically, Table A confirms this: **53.2%
> of retrieved experiences come from the late stage** (most on-policy), while only
> **27.0% come from the early stage**, demonstrating that our design effectively
> prioritizes the most up-to-date experiences without any explicit recency mechanism.
> Furthermore, as the bank reaches capacity, the oldest entries are evicted (lines 163-164 right), further ensuring that severely outdated experiences do not accumulate. To
> validate the importance of this alignment, we conduct a controlled mismatch experiment
> where experience embeddings are produced by a frozen zero-shot base model while query embeddings still come from the trained policy. This causes consistent performance
> drops of **2.65%, 2.85%, and 3.14%** on ExpBench-Generic, ExpBench-Sci, and
> ExpBench-Rec respectively, confirming that retrieval quality critically depends on
> query-experience embedding alignment. Together, these results demonstrate that
> ExpWeaver achieves robust experience management through current-policy encoding, which simultaneously ensures alignment for new experiences and naturally deprioritizes
> older ones via embedding compatibility.
>
> **Table A. Proportion of retrieved experiences from each training stage.**
>
> | Stage                           | Retrieval Proportion |
> |---------------------------------|----------------------|
> | Early (most off-policy)         | 27.0%               |
> | Mid (between early and late)    | 19.8%               |
> | Late (most on-policy)           | 53.2%               |
>
> ---
>
> **[W4: Notation Consistency on $e$]** $s_i$ in Algorithm 1 refers to the textual summary stated in lines 142-143 left, not the full experience $e$. We will clarify the notation to make this distinction explicit.
>
> ---
>
> **[Q1: Experience Impact on Performance]** To quantify the impact of off-policy experiences, we conduct a memory window ablation by restricting the experience bank to different training stages: **Early** (first 1/3, most off-policy), **Mid** (middle 1/3), **Late** (final 1/3, most on-policy), **Last 50%**, and **Full**. Results are reported as relative improvement over the early-only baseline. As shown in Table A, the gap between early-only and late-only is moderate (**+2.47%**), indicating that while on-policy experiences are more beneficial, off-policy experiences still contribute meaningful value. The full memory bank achieves the best performance (**+4.59%**), showing that older experiences provide complementary coverage beyond on-policy ones. These results demonstrate that ExpWeaver effectively manages off-policy distortion via current-policy encoding and FIFO eviction.
>
> **Table A. Relative performance improvement over early-only baseline.**
>
> | Memory Source | Relative Gain |
> |---------------|---------------|
> | Early only| baseline |
> | Mid only | +1.83% |
> | Late only | +2.47% |
> | Last 50% | +3.56% |
> | Full | **+4.59%** |

---

> > ### Author Rebuttal · Reviewer_gML7 · 2026-04-01
> >
> > The reviewer thanks the authors for their response and clarification.
> > In addition, the reviewer encourages the authors to update the manuscript accordingly.
> > This reviewer would like to increase the scoring.

---

> > > ### Author Response · Authors · 2026-04-01
> > >
> > > We sincerely thank the reviewer for the positive feedback and encouragement. We will carefully incorporate the clarifications and suggested updates into the revised manuscript to improve its clarity and completeness. We also greatly appreciate the reviewer’s willingness to increase the score.

---

### Official Review · Reviewer_saZB · 2026-03-11

**Soundness:** 2
**Presentation:** 3
**Significance:** 4
**Originality:** 3
**Overall Recommendation:** 5
**Confidence:** 4

**Summary:**

ExpWeaver is a framework for experience-augmented LLM agents that replaces text-based RAG with a latent representation approach.
Past experiences are encoded as hidden states, retrieved at each decoding step via cross-attention, and trained end-to-end with GRPO.
Evaluation on the authors' benchmark ExpBench across 13 tasks shows best performance on 12/13 tasks, with better token efficiency than text-based RAG baselines.

**Compliance With Llm Reviewing Policy:**

Affirmed.

**Final Justification:**

Accept. Th rebuttal have addressed my main concerns with newly added experiments and I decide to raise my assessment.

**Key Questions For Authors:**

Please see weaknesses

**Limitations:**

1. **Restricted Model Scale**
All experiments are conducted on Qwen2.5-3B-Instruct only. It remains unclear whether the latent retrieval advantage persists on larger models (e.g. 7B) where the LLM's internal knowledge is already richer, or across different architectures.

2.**Experience Summarization Is Not End-to-End**
The text summaries fed into the experience bank are generated by a fixed prompt template before encoding. This summarization step is decoupled from the end-to-end RL optimization

**Strengths And Weaknesses:**

## Summary of Strengths
- **Broad task coverage.** The 13 tasks span QA, reasoning, coding, AI4Sci, and recommendation, giving relatively broad empirical coverage.

## Summary of Weaknesses
My weaknesses are ordered by importance (high → low).

1. **Self-designed benchmark ExpBench.** The paper heavily relies on ExpBench and reports SOTA on 12/13 tasks. Without independent validation, it is hard to rule out benchmark-level bias (task design, experience construction protocol, metric choices) in favor of the proposed method.

2. **Computational cost of per-step latent retrieval.** The method applies cross-attention over the experience bank at every decoding step, which can introduce substantial cost scaling with sequence length × bank size. The paper reports token efficiency but lacks wall-clock/FLOPs/memory comparisons.

3. **Lack of relevant trainable baselines.** If search-agent methods are counted as experience-learning baselines, then more directly relevant memory-management methods should also be included and discussed (e.g., MemGen-style latent-space approaches).

---

> ### Author Rebuttal · Authors · 2026-03-31
>
> Thanks for the constructive comments. We address each point in detail below and will include these results in the revised paper.
>
> **[W1: Benchmark Design]** We clarify that **ExpBench is not a newly introduced benchmark, but a curated collection of well-established, publicly available benchmarks** with independent origins and broad adoption. As described in **lines 228–265 and Appendix D**, ExpBench spans multiple task domains (e.g., question answering, reasoning, coding, scientific applications, and recommendation), and none of the constituent tasks are designed by the authors. All evaluation metrics are directly inherited from the original benchmarks, and the experience construction follows an established and reproducible protocol from prior work (Ouyang et al., 2025). Thus, ExpBench does **not introduce new tasks, datasets, or metrics**, but **unifies diverse existing benchmarks under a common experience-learning framework**, reducing benchmark-level bias through independently validated sources. We will release full benchmark construction details, including data splits and preprocessing, in the final version.
>
> ---
>
> **[W2: Computational Cost and Efficiency]** **1. Inference Latency.** We refer the reviewer to our response to Reviewer rVqs (W1&Q1) for detailed wall-clock comparisons. In summary, ExpWeaver is only **30–35%** slower than R1 while being about **45% faster than Search-R1**, demonstrating a favorable efficiency–performance trade-off. **2. Training Memory.** Table A shows GPU memory (settings in Appendix A): ExpWeaver uses the **least memory across all scenarios**, lower than both Search-R1 and R1, indicating no additional memory burden from per-step latent retrieval. It also achieves better performance with significantly lower latency than Search-R1.
>
> **Table A. GPU memory comparison.**
>
> | Scenario           | R1      | Search-R1 | ExpWeaver |
> |--------------------|---------|-----------|-----------|
> | ExpBench-Generic   | 56.6 GB | 62.6 GB   | 41.2 GB   |
> | ExpBench-Sci       | 57.2 GB | 63.4 GB   | 44.9 GB   |
> | ExpBench-Rec       | 58.7 GB | 64.8 GB   | 47.5 GB   |
>
> ---
>
> **[W3: New relevant trainable baseline comparison]** We note that **Section 4 already compares ExpWeaver against multiple strong SOTA trainable baselines**. Following your suggestion, we additionally include MemGen (Zhang et al., 2025) in Table A, and ExpWeaver consistently outperforms it across all tasks. While both operate in latent space, two key differences explain the gap: **(1)** MemGen uses a trigger to decide **when** to access memory, whereas ExpWeaver retrieves at **every decoding step** using the current hidden state, enabling finer-grained integration; **(2)** MemGen relies on parametric memory from model weights, while ExpWeaver retrieves from an **external experience bank**, providing richer and more diverse patterns. The particularly large gain on **ExpBench-Rec** further indicates stronger transferability in ranking scenarios.
>
> **Table A. Performance comparison with MemGen (Qwen2.5-3B).**
>
> | Method     | ARC-C | CSQA  | GPQA  | MMLU  | OBQA  | GSM8K | GSM-Sym | MATH  | HumanEval+ | MBPP+ | Chem-TDC | Movie MRR | Music MRR |
> |------------|-------|-------|-------|-------|-------|-------|---------|-------|------------|-------|----------|-----------|-----------|
> | MemGen     | 81.33 | 78.44 | 26.67 | 66.89 | 74.22 | 85.33 | 81.56   | 64.11 | 71.28      | 72.63 | 64.89    | 22.17     | 18.84     |
> | ExpWeaver  | **84.00** | **82.00** | **35.00** | **68.00** | **80.00** | **89.78** | **85.00** | **65.33** | **72.00** | **78.00** | **69.58** | **39.55** | **33.21** |
>
> ---
>
> **[Q1: Model Scale + Architecture Generalization]** Please refer to our response to Reviewer rVqs (W3&Q3).
>
> ---
>
> **[Q2: Experience Summarization]** We acknowledge that the summarization step uses a fixed prompt template, but this does **not** undermine the end-to-end nature of ExpWeaver. The optimization is performed on the **latent encoding**, not the text itself: the summary is only an intermediate representation, while its latent embedding is produced by the **current policy** and continuously updated during RL training. As the policy evolves, the same summary can map to different latent embeddings, preserving the on-policy alignment described in Section 3.1. All downstream components, including retrieval similarity, cross-attention aggregation, and gated integration, are jointly optimized end-to-end through RL. Making the pipeline fully differentiable through summarization remains an **open challenge**, as it requires backpropagation through non-differentiable discrete text generation and is commonly handled similarly in prior work (Ouyang et al., 2025; Zhao et al., 2024; Wang et al., 2024). Finally, all baselines use the same fixed summarization protocol, yet ExpWeaver still achieves **>6.8% average improvement over the strongest baseline**, indicating that the gains stem from the latent integration mechanism rather than summarization quality.

---

> > ### Author Rebuttal · Reviewer_saZB · 2026-04-03
> >
> > Thank the authors for addressing my concerns and for the newly added MemGen baseline, which strengthens the experimental evaluation of this paper. I raise my score to 5.

---

> > > ### Author Response · Authors · 2026-04-03
> > >
> > > We sincerely thank the reviewer for the positive feedback and encouragement. We will carefully incorporate the clarifications and suggested updates into the revised manuscript to further enhance its clarity and completeness. We also greatly appreciate the reviewer’s willingness to consider increasing the score.

---

### Official Review · Reviewer_rVqs · 2026-03-21

**Soundness:** 3
**Presentation:** 3
**Significance:** 3
**Originality:** 3
**Overall Recommendation:** 4
**Confidence:** 4

**Summary:**

This paper aims to enhance LLM agent's planning and reasoning capability by leveraging past interactions as experience. Different from existing methods, it proposes to reuse the experience in the latent space by designing a latent RAG algorithm, namely ExpWeaver. ExpWeaver maintains an experience bank with hidden representations of reusable trajectories encoded by the policy model, performs the latent retrieval at each decoding step to select useful experience, and integrates these latent representations into the policy through a gated mechanism. ExpWeaver is trained jointly with the original policy model. Extensive experiments are conducted to demonstrate the effectiveness of ExpWeaver in enhancing solving complex reasoning tasks.

**Compliance With Llm Reviewing Policy:**

Affirmed.

**Final Justification:**

Weakly Accept.
The authors have clarified my concerns. The proposed method have made significant improvements over baseline methods, even though some of its properties, such as the computational cost and llm-specific training, make it less attractive. I will adjust my scoring accordingly.

**Key Questions For Authors:**

1. The efficiency of ExpWeaver is not evaluated. ExpWeaver performs retrieval task in each decoding step, which can significantly increase the inference latency. What is the inference efficiency of ExpWeaver compared with existing methods? How the size of the trajectory corpus affect the efficiency?
2. The proposed method is not LLM agnostic, and needs to train jointly with LLM based on the paper, which makes it less attractive. Can it be generalize to different LLMs once trained?
3. The experiments are only conducted on Qwen. What if using different LLMs?

**Limitations:**

Yes.

**Strengths And Weaknesses:**

# Strengths

1. The proposed method is an interesting and novel practice in leveraging historical trajectories for enhancing reasoning capability of LLMs.
2. The paper provides detailed analysis of the experimental results.
3. The proposed method is clearly described with figures.

# Weaknesses
1. The efficiency of ExpWeaver is not evaluated. ExpWeaver performs retrieval task in each decoding step, which can significantly increase the inference latency. What is the inference efficiency of ExpWeaver compared with existing methods? How the size of the trajectory corpus affect the efficiency?
2. The proposed method is not LLM agnostic, and needs to train jointly with LLM, which makes it less attractive.
3. The experiments are only conducted on Qwen. What if using different LLMs?

---

> ### Author Rebuttal · Authors · 2026-03-31
>
> Thanks for the constructive comments. We address each point in detail below and will include these results in the revised paper.
>
> **[W1&Q1: Inference Efficiency and Scalability with Corpus Size]** **1. Inference latency comparison.**  Table A reports the average inference time per sample across three scenarios. ExpWeaver is only **30–35%** slower than R1 despite retrieval at every decoding step, because FAISS approximate nearest neighbor search adds sub-millisecond latency per step. It remains about **45% faster than Search-R1** across all scenarios while substantially outperforming it in accuracy, demonstrating a strong efficiency–performance trade-off. **2. Effect of corpus size.**  Table B shows that increasing the experience bank from **10% to 100%** of full capacity raises inference latency by only **2.1%**, indicating that ExpWeaver scales gracefully with corpus size.
>
> **Table A. Average inference time (s/sample).**
>
> | Scenario           | R1   | Search-R1 | ExpWeaver |
> |-------------------|------|-----------|-----------|
> | ExpBench-Generic  | 0.19 | 0.46      | 0.25      |
> | ExpBench-Sci      | 0.21 | 0.48      | 0.27      |
> | ExpBench-Rec      | 0.24 | 0.59      | 0.32      |
>
> **Table B. Inference latency of ExpWeaver with corpus size on ExpBench-Generic.**
>
> | Corpus Size | Relative Latency Change |
> |-------------|------------------------|
> | 10% | baseline |
> | 30% | +0.4% |
> | 50% | +0.7% |
> | 70% | +1.2% |
> | 90% | +1.8% |
> | 100% | +2.1% |
>
> ---
>
> **[W2&Q2: LLM-Agnosticism]** We would like to clarify that joint training with the LLM is a **deliberate design choice**, not a limitation. The core insight of ExpWeaver is that on-policy alignment between experience embeddings and query hidden states is essential for effective latent retrieval as stated in the Introduction and Preliminaries. Decoupling retrieval from the LLM — as in LLM-agnostic baselines (Ouyang et al., 2025; Zhao et al., 2024; Wang et al., 2024) — reintroduces the very bottlenecks (semantic mismatch, separate optimization objectives) identified in recent works (Lin et al., 2025; Feng et al., 2025; Trivedi et al., 2023; Li et al., 2025) that ExpWeaver aims to resolve. Empirically, ExpWeaver outperforms the best LLM-agnostic baseline (LLM-R) by **14.73%** on ExpBench-Generic, demonstrating the cost of decoupling retrieval from generation. The additional parameters $\psi_{exp}$ are minimal, and we adopt parameter-efficient LoRA fine-tuning, making adaptation to new LLMs lightweight compared to full pre-training. Moreover, strong baselines like Search-R1 also require RL-based LLM fine-tuning. ExpWeaver further shows strong transferability, achieving **16.32%** zero-shot and **15.21%** few-shot improvements over Search-R1 across domains.
>
> ---
>
> **[W3&Q3: Experiments on other LLMs]**  We conducted additional experiments on Llama3.2-3B-Instruct and Qwen2.5-7B-Instruct in Tables A and B. ExpWeaver consistently outperforms both R1 and Search-R1, the strongest baselines, across all task categories on both LLM backbones, demonstrating that the latent experience integration mechanism generalizes well beyond the Qwen2.5-3B setting. Notably, the performance gains are particularly pronounced on larger models (Qwen2.5-7B), suggesting that ExpWeaver benefits from stronger backbone representations. The results of other baselines on the two models can be found [here](https://anonymous.4open.science/r/ExpWeaver).
>
> **Table A. Performance comparison on Llama3.2-3B-Instruct.**
>
> | Method     | ARC-C | CSQA  | GPQA  | MMLU  | OBQA  | GSM8K | GSM-Sym | MATH  | HumanEval+ | MBPP+ | Chem-TDC | Movie MRR | Music MRR |
> |------------|-------|-------|-------|-------|-------|-------|---------|-------|------------|-------|----------|-----------|-----------|
> | R1         | 70.20 | 66.52 | 16.67 | 61.11 | 64.92 | 78.22 | 71.78   | 55.10 | 64.10      | 66.25 | 57.56    | 16.83     | 13.47     |
> | Search-R1  | 73.74 | 70.11 | 20.00 | 64.22 | 68.34 | 76.67 | 70.00   | 53.51 | 64.10      | 65.00 | 59.33    | 18.33     | 22.12     |
> | ExpWeaver  | **78.28** | **75.56** | **28.33** | **65.33** | **74.67** | **82.00** | **76.89** | **59.33** | **65.38** | **73.75** | **64.67** | **35.72** | **27.84** |
>
> **Table B. Performance comparison on Qwen2.5-7B-Instruct.**
>
> | Method     | ARC-C | CSQA  | GPQA  | MMLU  | OBQA  | GSM8K | GSM-Sym | MATH  | HumanEval+ | MBPP+ | Chem-TDC | Movie MRR | Music MRR |
> |------------|-------|-------|-------|-------|-------|-------|---------|-------|------------|-------|----------|-----------|-----------|
> | R1         | 90.15 | 86.52 | 30.00 | 77.33 | 84.51 | 94.22 | 90.22   | 75.06 | 79.49      | 77.50 | 65.11    | 20.94     | 17.83     |
> | Search-R1  | 91.92 | 88.31 | 31.67 | 78.67 | 86.33 | 92.89 | 88.67   | 73.70 | 79.49      | 76.25 | 67.78    | 22.27     | 27.58     |
> | ExpWeaver  | **94.44** | **92.22** | **45.00** | **80.22** | **91.56** | **96.67** | **95.33** | **79.56** | **80.77** | **85.00** | **73.33** | **44.17** | **36.44** |

---

> > ### Author Rebuttal · Reviewer_rVqs · 2026-04-02
> >
> > Thanks for the detailed response. The proposed method have made significant improvements over baseline methods, even though some of its properties make it less attractive. I will adjust my scoring accordingly.

---

> > > ### Author Response · Authors · 2026-04-02
> > >
> > > We sincerely thank the reviewer for the thoughtful consideration and for recognizing the empirical improvements of our method over strong baselines. We appreciate the reviewer’s willingness to re-evaluate the score.

---

### Decision · Program_Chairs · 2026-04-30

**Decision:**

Accept (regular)

**Comment:**

This work studies experience retrieval for enhancing LLMs' reasoning and planning. The authors propose to encode, retrieve, and integrate relevant experience in the latent space. The idea was demonstrated to be useful across 13 QA tasks. Reviewers generically recognized the technical novelty and wide empirical gains.

Several concerns were raised by reviewers about the scalability, generalization to different LLMs, and the off-policy drift from the offline experience bank. The authors present experiments to address those concerns directly and clearly, as acknowledged by all reviewers, as well. Therefore, I join in the reviewers to recommend the acceptance of this work.